# Can an AI Agent Safely Run a Government?
# Existence of Probably Approximately Aligned Policies

**Frédéric Berdoz**
ETH Zürich
fberdoz@ethz.ch

**Roger Wattenhofer**
ETH Zürich
wattenhofer@ethz.ch

## Abstract

While autonomous agents often surpass humans in their ability to handle vast and complex data, their potential misalignment (i.e., lack of transparency regarding their true objective) has thus far hindered their use in critical applications such as social decision processes. More importantly, existing alignment methods provide no formal guarantees on the safety of such models. Drawing from utility and social choice theory, we provide a novel quantitative definition of alignment in the context of social decision-making. Building on this definition, we introduce *probably approximately aligned* (i.e., near-optimal) policies, and we derive a sufficient condition for their existence. Lastly, recognizing the practical difficulty of satisfying this condition, we introduce the relaxed concept of *safe* (i.e., nondestructive) policies, and we propose a simple yet robust method to safeguard the black-box policy of any autonomous agent, ensuring all its actions are verifiably safe for the society.

## 1 Introduction

The deployment of AI systems in critical applications, such as social decision-making, is often stalled by the following two shortcomings: 1) They are *brittle* and usually provide no guarantees on their expected performance when deployed in the real world [8], and 2) there is no formal guarantee that the objective they have been trained against, typically a scalar quantity such as a *loss* or a *reward*, faithfully represents human interest at large [41]. Addressing these limitations is commonly referred to as *AI alignment*, an umbrella term including a wide array of methods supposed to make AI systems of different modalities behave as intended [17, 24].

Yet, to our knowledge, every metric for alignment is *a posteriori*, i.e., a system is deemed aligned as long as it does not display misaligned behavior (e.g., through red teaming [14]). This stems from the fact that most of these methods focus on aligning generative models of complex modalities (text, images, video, audio, etc.) where the input and output domains are particularly vast, and where no single metric can perfectly represent the intended behavior.

In the context of critical (e.g., social) decision processes, where an autonomous agent must repeatedly take actions in a complex environment with many stakeholders, *a posteriori* alignment is not sufficient. Indeed, for the same reasons that a society would not trust a human policymaker with hidden motives and unknown track record, it would also distrust an autonomous policymaker whose objective is not, *a priori*, perfectly clear and verifiably aligned, as the cost of a single bad action (due to their known *brittleness*) could easily outweigh the benefits of leveraging such systems. This issue is accentuated by the fact that, unlike humans, holding a deceptive AI agent accountable remains a challenge [25].

Conversely, recent breakthroughs in AI have significantly increased its potential for beneficial use in these critical settings. For example, tax rates and public spending are typically set periodically by a parliament. However, this small group of representatives is inevitably overwhelmed by the vast amount of complex economic data as well as the pleas of millions of individuals. Due to this

38th Conference on Neural Information Processing Systems (NeurIPS 2024).

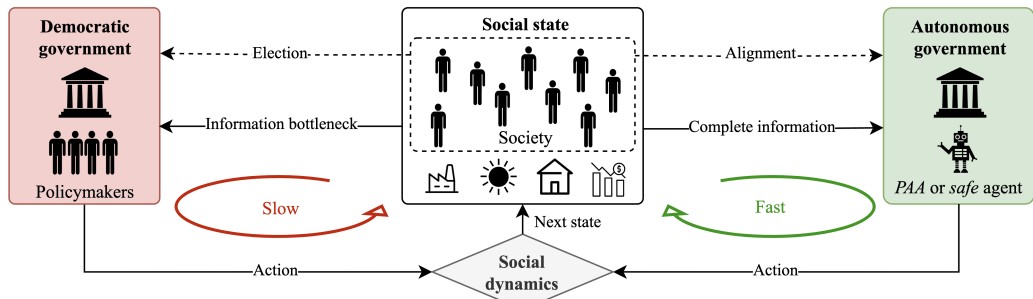

Figure 1: Democratic (left) vs autonomous (right) governments. Transparent elections must be replaced with reliable alignment mechanisms.

information bottleneck, governments can only make educated guesses about what is best for the society (assuming they can come to an agreement in the first place). On the other hand, provided that it is verifiably aligned, an autonomous government could efficiently leverage this vast amount of information in order to find the optimal tax rates and public spending (see Figure 1).

While it is clear that *a priori* alignment is essential for the safe delegation of such decision power, it comes at the cost of two unavoidable prerequisites. Firstly, it requires a definition of alignment that is both quantitative and measurable. Given our focus on social decision processes, where the perception of each action may differ among individuals, we define and quantify alignment by drawing from long-established theories of utility and social choice. For completeness, we also discuss all the assumptions that allow such a metric to exist. Secondly, one can only ensure that an agent is *a priori* aligned if one independently understands (at least reasonably well) how its available actions may affect society, regardless of how those effects are perceived. Indeed, asserting the safety of an autonomous agent would be impossible if it could take actions with unknown consequences. Although we make no assumption of the exact nature of this knowledge (domain specific expertise, heuristics, physical/data-driven modeling, or a mix thereof), we assume it is encapsulated in a world model that estimates, given an action and a current state, the probability of moving to any other state.

A natural question then arises: Can we leverage this knowledge to construct a verifiably aligned policy? Or, at the very least, can we use it to ensure the safety of a more complex (black-box) policy? We address these questions by introducing *probably approximately aligned* (PAA) and *safe* policies, and by studying their existence based on the quality of this knowledge. Akin to the *probably approximately correct* (PAC) framework in the theory of learnability [38], we are also interested in the sample complexity of finding PAA policies. In our setting, this complexity is two-faceted: First, the number of calls to the world model (similar to the number of calls to a perfect generative model in [18]), and second, the amount of feedback (think ballots) required to confidently rank which state is socially preferred. While we are not yet concerned with *efficient* PAA policies, whose sample complexity is at most polynomial with respect to the desired tolerances, we argue that these complexities should, at least, be independent of the number of possible states (similarly to sparse sampling algorithms [19]), as most natural state spaces are infinite. We refer to such policies as *computable*. Conversely, as one can decide which decisions are delegated to autonomous agents, we assume that the action space is finite. Concretely, our contribution is threefold:

- First, we define a new type of Social Markov Decision Processes, replacing the traditional reward with aggregated utilities of individuals. Expanding on this definition, **we present a formal quantitative definition of alignment** in the context of social choice, which naturally leads to the concept of *probably approximately aligned* (PAA) policies.

- Next, given an approximate world model with sufficient statistical accuracy (which we quantitatively derive), **we prove the existence of PAA policies** by amending a simple sparse sampling algorithm.

- Finally, **we propose a simple and intuitive method to safeguard any black-box (potentially misaligned) policy** in order to ensure that it does not take any destructive decisions, i.e., actions that might lead to a state where even the optimal policy is unsatisfactory. We refer to these adapted policies as *safe* policies.

## 2  Background

### 2.1  Utility and social choice theory

We are interested in building autonomous agents whose objective is to maximizes social satisfaction by taking actions that alter the state of the society. In this section, we define what is meant by *social satisfaction* (often called *social welfare*), and we provide the conditions under which it is quantifiable.

#### 2.1.1  Utility theory

It is commonly assumed that the agency of an individual is governed by its internal binary preference relationship $\preceq$ over the set of outcomes $\mathcal{S}$. When presented with two choices $s$ and $s'$, the individual will introspect its satisfaction (welfare) levels and either strictly prefer one outcome ($s \prec s'$ or $s' \prec s$) or be indifferent to both ($s \sim s'$). We are interested in quantitatively measuring these welfare levels. Debreu's representation theorem [10] states that if this preference relationship is complete, continuous, and transitive on the topological space $\mathcal{S}$, then there exists infinitely many continuous, real-valued functions $u : \mathcal{S} \rightarrow \mathbb{R}$ (called *utility functions*) such that $u(s) \leq u(s') \Leftrightarrow s \preceq s'$, $\forall s, s' \in \mathcal{S}$ (note that any strictly monotonically increasing function can transform a valid utility function into another). While these findings establish the existence of these utility functions, which are proxies for the intrinsic welfare levels of individuals, they do not provide insights into their measurability and interpersonal comparability. It is these two properties, however, that eventually determine what measure of social welfare can be derived. In a nutshell, measurability and comparability impose how much information can be extracted from the values $|u_i(s) - u_i(s')|$ and $u_i(s) - u_j(s)$, respectively, for any $i \neq j$ and $s \neq s'$. We detail the various measurability and comparability levels in Appendix A.1.1, and we refer to these levels as the *informational basis* of utilities. Apart from that, we fix $0 < U_{min} \leq u_i(s) \leq U_{max} < \infty$ for any $s$ and $i$ (we will allow $U_{min} = 0$ in specific cases). That is, we assume that individuals cannot be infinitely satisfied or dissatisfied, and that they must scale their utilities when reporting (which does not imply measurability or comparability!). Lastly, we define $\Delta U \triangleq U_{max} - U_{min}$ and $\mathcal{U} \triangleq \{u : \mathcal{S} \rightarrow [U_{min}, U_{max}]\}$.

#### 2.1.2  Social choice theory

Let $\mathcal{I}$ be a society composed of $N$ members, each with its preference relationship $\preceq_i$ and a corresponding utility function $u_i \in \mathcal{U}$ over state space $\mathcal{S}$, $i \in \mathcal{I}$. Let $\mathcal{R}_{\mathcal{S}}$ be the set of complete orderings on $\mathcal{S}$ and $\mathbf{u} \in \mathcal{U}^N$ be a vector gathering the utility functions of all individuals. A social welfare functional $f$ (SWFL) is a mapping $\mathcal{D}_f \rightarrow \mathcal{R}_{\mathcal{S}}$ with $\mathcal{D}_f \subseteq \mathcal{U}^N$. In other words, it is an aggregator of individuals' utilities, indirectly preferences. A long line of work [9, 32, 29] has attempted to define which conditions this SWFL should satisfy (sometimes called axioms of cardinal welfarism, see Appendix A.1.2 for an extensive list of these properties and their respective implications). For the remainder of this work, we will follow the common assumption that any reasonable SWFL should satisfy the following: universality (U), informational basis invariance (XI), independence of irrelevant alternatives (IIA), weak Pareto criterion (WP) and anonymity (A).

An important result [29] states that, for any informational basis (X) listed in Appendix A.1.1 and any SWFL $f$ satisfying (XI), (U), (IIA) and (WP), there exists a social welfare function (SWF) $W : \mathbb{R}^N \rightarrow \mathbb{R}$ such that, if $W(\mathbf{u}(s)) > W(\mathbf{u}(s'))$, then $s$ ranks strictly higher than $s'$ in $f(\mathbf{u})$. This is important as it states that the best social state must maximize a certain function $W$, which can therefore be used as a measure of social satisfaction. In other words, the non-welfare characteristics (i.e., any information influencing $f(\mathbf{u})$ beside $\mathbf{u}$, such as the judgement of an AI agent) are of secondary importance, as they can only break ties between $s$ and $s'$ such that $W(\mathbf{u}(s)) = W(\mathbf{u}(s'))$ and cannot be detrimental to the society. Although we do not require it in this work, maximization of $W$ can be made sufficient if one imposes *Welfarism* (W), e.g., by replacing (WP) with *Weak Pareto Indifference* (WPI) or more drastically by imposing *Continuity* (C) (see Appendix A.1.2 for more details). We are left with the following question: Given a SWFL satisfying (XI), (U), (IIA), (WP) and (A), what is the form of the corresponding SWF? It turns out that the choice is relatively limited and depends mostly on the informational basis invariance (XI). It has been shown, with additional small technical assumptions [7], that the power mean defined in Eq. (1) covers all possible SFWL. See Appendix A.1.3 for a detailed mapping between informational bases, SWFLs and parameter $q$.

$$W_q(\mathbf{u}(s); \mathcal{I}) = \begin{cases} \min_{i \in \mathcal{I}} u_i(s) & q = -\infty \\ \sqrt[q]{\frac{1}{|\mathcal{I}|} \sum_{i \in \mathcal{I}} u_i(s)^q} & q \in \mathbb{R}^* \\ \sqrt[|\mathcal{I}|]{\prod_{i \in \mathcal{I}} u_i(s)} & q = 0 \\ \max_{i \in \mathcal{I}} u_i(s) & q = \infty \end{cases} \tag{1}$$

### 2.1.3 Future discounted social welfare

At deployment, a safe autonomous agent must provide assurances that its future actions will continue to serve the best interests of society. This becomes ill-defined if these interests evolve with time. To address this, we assume that both $\mathcal{I}$ and $\mathbf{u}$ are constant, i.e., $u_i(s; t) = u_i(s; t') \equiv u_i(s)$ for all $s$, $i$ and discrete times $t \neq t'$. In addition, we also assume that the meaning of these utilities does not change with time, that is, if $u_i(s; t) \geq u_i(s'; t')$ for states $s, s'$ and times $t \neq t'$, then $i$'s welfare is at least as high in state $s$ at time $t$ than in state $s'$ at time $t'$ (or vice versa for $\leq$). Finally, we assume that the SWFL $f$ is such that its corresponding SWF remains the same. In other words, only the method to break ties between states can evolve through time. This makes it possible to predict, at time $t$, what will be the satisfaction levels at time $t' > t$ in any given state. However, to model the fact that humans prefer immediate reward, we discount the utility of the state at time $t'$ with a factor $\gamma^{(t'-t)}$ when comparing it with the utility of the state at time $t$, where $\gamma \in [0, 1[$ is a discount factor. From these assumptions, it becomes possible, at time $t = 0$, to quantify the cumulative social welfare of any future state trajectory $s_1 s_2 s_3 s_4 ...$ by computing the quantity $\sum_{t=0}^{\infty} \gamma^t W_q(\mathbf{u}(s_{t+1}))$, which we will refer to as the *future discounted social welfare* of that trajectory (see Section 2.2.2). Using this quantity, we formally define alignment as follows:

*An autonomous agent is aligned if and only if it always takes actions that maximize the expected future discounted social welfare.*

## 2.2 Social dynamics

The expectation in the above definition accounts for the inherent randomness of most natural systems. In this section, we model the social dynamics as a particular type of Markov Decision Process (MDP), where the probability of transitioning to any state depends solely on the current state and next action.

### 2.2.1 Markov decision process

Let $\mathcal{M} = (\mathcal{S}, \mathcal{A}, p, r, \gamma)$ be an infinite horizon, $\gamma$-discounted, discrete time MDP where $\mathcal{S}$ is the state space (discrete or continuous), $\mathcal{A}$ is the action space (discrete or continuous), $p : \mathcal{S} \times \mathcal{A} \to \mathcal{P}(\mathcal{S})$ is the transition dynamics of the environment (with $\mathcal{P}(\mathcal{S})$ the set of probability distributions over $\mathcal{S}$), $r : \mathcal{S} \times \mathcal{A} \to [R_{min}, R_{max}]$ is the reward of the environment, and $\gamma \in [0, 1[$ is a discount factor (favoring immediate over distant rewards). Given $s \in \mathcal{S}$ and $a \in \mathcal{A}$, $p(s'|s, a)$ is the probability of transitioning to state $s'$ after taking action $a$ in state $s$, and $r(s, a)$ is the expected immediate reward after taking that action. In MDPs, actions are chosen according to a policy $\pi : \mathcal{S} \to \mathcal{P}(\mathcal{A})$, with $\pi(a|s)$ the probability of taking action $a$ in state $s$. Given an initial state $s_0$, the tuple $(\mathcal{M}, \pi, s_0)$ fully defines a distribution $p_\tau$ over trajectories $\tau = s_0 a_0 s_1 a_1 s_2 a_2 ...$, where $a_t \sim \pi(\cdot|s_t)$ and $s_{t+1} \sim p(\cdot|s_t, a_t)$. If the environment dynamics or the policy are deterministic, we will use the slight abuse of notation $s_{t+1} = p(s_t, a_t)$ and $a_t = \pi(s_t)$, respectively. The efficacy of a given policy $\pi$ is measured by the state and state-action value functions, defined respectively as follows:

$$V^\pi(s) = \mathbb{E}_{\tau \sim p_\tau(\cdot|\pi, s_0 = s)} \left[ \sum_{t=0}^{\infty} \gamma^t r(s_t, a_t) \right], \qquad Q^\pi(s, a) = r(s, a) + \gamma \mathbb{E}_{s' \sim p(\cdot|a, s)}[V^\pi(s')].$$

For a given state $s$ and action $a$, the optimal state and action-state value functions are defined by $V^*(s) = \sup_\pi V^\pi(s)$ and $Q^*(s, a) = \sup_\pi Q^\pi(s, a)$. Given $\varepsilon \geq 0$, a policy is called $\varepsilon$-optimal (or optimal) if it satisfies $V^\pi(s) \geq V^*(s) - \varepsilon$ (respectively $V^\pi(s) = V^*(s)$) for all $s$. Given small technical assumptions, it can be shown that there always exists an optimal policy [37, 12].

### 2.2.2 Social Markov decision process

**Definition** (*Social Markov Decision Process*). Let $\mathcal{I}$ be a society with utility profile $\mathbf{u} \in \mathcal{U}^N$ and social welfare function $W_q$. In addition, let $\mathcal{S}$ and $\mathcal{A}$ be the corresponding state and action spaces, $p$ the social environment dynamics and $\gamma$ a discount factor. The MDP $(\mathcal{S}, \mathcal{A}, p, r_{\mathcal{I}}, \gamma)$ with reward $r_{\mathcal{I}}$ defined by

$$r_{\mathcal{I}}(s, a) = \mathbb{E}_{s' \sim p(\cdot|s,a)}[W_q(\mathbf{u}(s'); \mathcal{I})] \tag{2}$$

is a Social Markov Decision Process (SMDP), denoted $\mathcal{M}_{\mathcal{I}} = (\mathcal{S}, \mathcal{A}, p, W_q, \mathbf{u}, \gamma)$.

In this setting, $\mathcal{S}$ contains all the possible states of the $N$ individuals, as well as all those of the environment in which they evolve, and $\mathcal{A}$ contains all the actions that are delegated to an autonomous agent. Expanding on this definition, we can formally define the alignment metric proposed above:

**Definition** (*Expected Future Discounted Social Welfare*). Let $\mathcal{M}_{\mathcal{I}} = (\mathcal{S}, \mathcal{A}, p, W_q, \mathbf{u}, \gamma)$ be a SMDP. The expected future discounted social welfare of a policy $\pi$ in state $s$ is defined as

$$\mathcal{W}^{\pi}(s) = \mathbb{E}_{\tau \sim p_{\tau}(\cdot|\pi, s_0 = s)} \left[ \sum_{t=0}^{\infty} \gamma^t W_q(\mathbf{u}(s_{t+1}); \mathcal{I}) \right],$$

and takes value between $\mathcal{W}_{min} \triangleq \frac{U_{min}}{1-\gamma}$ and $\mathcal{W}_{max} \triangleq \frac{U_{max}}{1-\gamma}$, with $\Delta \mathcal{W} \triangleq \mathcal{W}_{max} - \mathcal{W}_{min}$.

As shown with the next lemma, the expected future discounted social welfare of a SMDP is equivalent to the state value function of the corresponding MDP. This equivalence makes it a natural metric for alignment, as it enables the use of a wide array of known results on MDPs.

**Lemma 1.** *For any SMDP $\mathcal{M}_{\mathcal{I}} = (\mathcal{S}, \mathcal{A}, p, W_q, \mathbf{u}, \gamma)$, the expected future discounted social welfare of a policy $\pi$ is the state value function of $\pi$ in the MDP $\mathcal{M} = (\mathcal{S}, \mathcal{A}, p, r_{\mathcal{I}}, \gamma)$, with $r_{\mathcal{I}}$ set in Eq. (2).*

The proof follows directly from the tower property of conditional expectations (see Appendix A.2.1).

### 2.2.3 Approximate rewards

If $p$ is unknown, the true reward of the SMDP in Eq. (2) can only be estimated *a posteriori*, i.e., after taking action $a$ in state $s$ multiple times and observing $W_q(\mathbf{u}(s'))$. This would require a long exploration phase if $\mathcal{S}$ is large, which can be costly and even impossible in critical decisions processes. Instead, one must usually plan using an approximate dynamics model $\hat{p} \in \mathcal{P}(\mathcal{S})$ to anticipate the effect of an action. Moreover, even if $p$ is known, computing $W_q(\mathbf{u}(s'))$ exactly for a given $s' \sim p$ would require full knowledge of $\mathbf{u}(s')$, which is only possible by obtaining feedback from the entire society about $s'$ without making additional assumptions on $\mathbf{u}$. For these reasons, we consider a more realistic scenario: Given a set of assessors $I_n \subseteq \mathcal{I}$ of size $n \leq N$ and an approximate dynamics model $\hat{p}$, the true reward can be approximated by asking the assessors about their utilities on the anticipated future societal states:

$$\hat{r}_{I_n}(s, a; K) = \hat{\mathbb{E}}_{s' \sim \hat{p}(\cdot|s,a)}^{K}[W_q(\mathbf{u}(s'); I_n)] \triangleq \frac{1}{K} \sum_{k=1}^{K} W_q(\mathbf{u}(s^k)). \tag{3}$$

where $\hat{\mathbb{E}}_{s' \sim \hat{p}}^{K}$ is a "Monte Carlo" estimation of $\mathbb{E}_{s' \sim \hat{p}}$, with $K \in \mathbb{N}$ samples independently drawn from $\hat{p}(\cdot|s,a)$, denoted $s^1, s^2 ..., s^K$. The core of our analysis is to understand how $K$, $n$ and the inaccuracies of $\hat{p}$ affect the validity of alignment guarantees.

## 3 Results

### 3.1 Existence of aligned policies

Having formally derived a quantitative measure of alignment $\mathcal{W}^{\pi}$ in the context of social decision processes, we are now prepared to introduce and prove the existence of verifiably aligned policies:

**Definition** (*Probably Approximately Aligned Policy*). Given $0 \leq \delta < 1$, $\varepsilon > 0$ and a SMDP $\mathcal{M}_{\mathcal{I}} = (\mathcal{S}, \mathcal{A}, p, W_q, \mathbf{u}, \gamma)$, a policy $\pi$ is $\delta$-$\varepsilon$-PAA (Probably Approximately Aligned) if, for any given $s \in \mathcal{S}$, the following inequality holds with probability at least $1 - \delta$:

$$\mathcal{W}^{\pi}(s) \geq \max_{\pi'} \mathcal{W}^{\pi'}(s) - \varepsilon \tag{4}$$

**Definition** (*Approximately Aligned Policy*). Given $\varepsilon > 0$, a policy $\pi$ is $\varepsilon$-AA (Approximately Aligned) if and only if it is $0$-$\varepsilon$-PAA.

We state below one of our main contribution, i.e., the existence of computable PAA and AA policies, which follows directly from Theorem 3 ($\delta > 0$) and Corollary 4 ($\delta = 0$) in the next section.

**Theorem 2** (Existence of PAA and AA policies). *Given a SMDP $\mathcal{M}_{\mathcal{I}} = (\mathcal{S}, \mathcal{A}, p, W_q, \mathbf{u}, \gamma)$ with $q \in \mathbb{R}$ and any tolerances $\varepsilon > 0$ and $0 \leq \delta < 1$, if there exists an approximate world model $\hat{p}$ such that*

$$\sup_{(s,a) \in \mathcal{S} \times \mathcal{A}} D_{KL}(p(\cdot|s,a)\|\hat{p}(\cdot|s,a)) < \frac{\varepsilon^2(1-\gamma)^4}{8\Delta \mathcal{W}^2}, \tag{5}$$

*then there exists a computable $\delta$-$\varepsilon$-PAA policy. Consequently, there also exists a computable $\varepsilon$-AA policy.*

## 3.2 Near optimal planning

We prove Theorem 2 by providing a planning policy $\pi_{PAA}$ and by proving it satisfies Eq. (4) under the given assumptions. To this end, we present a modified version of the sparse sampling algorithm [19] (which originally assumes that $p$ and $r$ are known, which is not the case here). Given some parameters $K, C$ and $n$, we define the recursive functions:

$$\hat{Q}^h(s,a;K,C,I_n) = \begin{cases} 0 & h = 0 \\ \hat{r}_{I_n}(s,a;K) + \gamma \hat{\mathbb{E}}^C_{s' \sim \hat{p}(\cdot|s,a)}\left[\hat{V}^{h-1}(s';K,C,I_n)\right] & h \in \mathbb{N}^* \end{cases} \tag{6}$$
$$\hat{V}^h(s;K,C,I_n) = \max_{a \in \mathcal{A}} \hat{Q}^h(s,a;K,C,I_n).$$

Intuitively, $\hat{Q}^h$ and $\hat{V}^h$ are recursive approximations of $Q^*$ and $V^*$, $K$ and $C$ controls the accuracy of the empirical expectation operators in $\hat{Q}$, $h$ controls how far one looks into the future, and $n$ controls the accuracy of the social welfare function estimates. The proposed PAA policy is simply the greedy policy acting on the state-action value estimates, i.e.,

$$\pi_{PAA}(s) = \max_{a \in \mathcal{A}} \hat{Q}^H(s,a;K,C,I_n). \tag{7}$$

It is deterministic in the sense that, for a given $\hat{Q}^H$, it outputs a single action. However, $\hat{Q}^H$ is non-deterministic since $I_n$ and $s'$ are sampled randomly. The next results clarifies under which conditions $\pi_{PAA}$ is indeed $\delta$-$\varepsilon$-PAA (Theorem 3) or $\varepsilon$-AA (Corollary 4).

**Theorem 3** ($\pi_{PAA}$ is $\delta$-$\varepsilon$-PAA). *Let $\mathcal{M}_{\mathcal{I}} = (\mathcal{S}, \mathcal{A}, p, W_q, \mathbf{u}, \gamma)$ be a SMDP with $q \in \mathbb{R}$ and $\hat{p}$ an approximate dynamics model such that $d \triangleq \sup_{(s,a)} D_{KL}(p(\cdot|s,a)\|\hat{p}(\cdot|s,a)) < \frac{\varepsilon^2(1-\gamma)^6}{8\Delta U^2}$ for any desired tolerances $\varepsilon > 0$ and $0 < \delta < 1$. For any $k \geq \log_\gamma\left(\frac{(1-\gamma)\varepsilon}{\Delta U} - \frac{\sqrt{8d}}{(1-\gamma)^2}\right)$, define $\beta \triangleq \left(\frac{(1-\gamma)^2\varepsilon}{8} - \frac{\sqrt{d}\Delta U}{\sqrt{8}(1-\gamma)} - \frac{(1-\gamma)\gamma^k \Delta U}{8}\right)$ and let $\pi_{PAA}$ be the policy defined in Eq. (7) with parameters*

- $H \geq \max\left\{1, \log_\gamma\left(\frac{\beta}{\Delta U}\right)\right\}$,

- $K \geq \frac{\Delta U^2}{\beta^2}\left((H-1)\ln\left(\sqrt[H-1]{24k}(H-1)|\mathcal{A}|\frac{\Delta U^2}{\beta^2}\right) + \ln\left(\frac{1}{\delta}\right)\right)$,

- $C \geq \frac{\gamma^2}{(1-\gamma)^2}K$,

- $n \geq N\left(1 + \frac{\Delta U^2 N}{2K}\Gamma\left(\beta, U_{min}, U_{max}, q\right)\right)^{-1}$,

*and where $\Gamma\left(\beta, U_{min}, U_{max}, q\right)$ is a function defined in Eq. (9). Then $\pi_{PAA}$ is a $\delta$-$\varepsilon$-PAA policy.*

**Corollary 4.** *Let $\mathcal{M}_{\mathcal{I}} = (\mathcal{S}, \mathcal{A}, p, W_q, \mathbf{u}, \gamma)$ be a SMDP with $q \in \mathbb{R}$ and $\hat{p}$ an approximate dynamics model such that $d \triangleq \sup_{(s,a)} D_{KL}(p(\cdot|s,a)\|\hat{p}(\cdot|s,a)) < \frac{\varepsilon^2(1-\gamma)^6}{8\Delta U^2}$ for any desired tolerance $\varepsilon > 0$.*

*Define $\beta \triangleq \frac{(1-\gamma)^2\varepsilon}{10} - \frac{\sqrt{2d}\Delta U}{5(1-\gamma)}$ and let $\pi_{PAA}$ be the policy defined in Eq. (7) with parameters $H, C$ and $n$ as in Theorem 3 and $K \geq \frac{\Delta U^2}{\beta^2}\left((H-1)\ln\left(\sqrt[H-1]{12}(H-1)|\mathcal{A}|\frac{\Delta U^2}{\beta^2}\right) + \ln\left(\frac{\Delta U}{\beta}\right)\right)$. Then $\pi_{PAA}$ is an $\varepsilon$-AA policy.*

See Appendix A.2.2 for the full derivation of these two results. We now outline a proof sketch. The skeleton of the proof is similar to the one proposed by Kearns et al. [19] for their original sparse sampling algorithm, although several additional tricks and intermediate results are necessary to accommodate the approximate world model $\hat{p}$ and reward $\hat{r}_\mathcal{I}$. First, we derive a concentration inequality for the power mean function in order to quantify the approximation error between $W_q(\cdot; I_n)$ and $W_q(\cdot; \mathcal{I})$. Using a slightly modified (two-sided) version of the Hoeffding-Serfling inequality [4, 33] (see Lemma 9 in Appendix A.2.1), we find the following bounds:

**Lemma 5.** *Let $W_q$ be the power-mean defined in Eq. (1), with $q \in \mathbb{R}$. Given $a, b \in \mathbb{R}_+^*$ (or $\mathbb{R}_+$ for $q = 1$) such that $a < b$, let $\mathcal{X} \in [a,b]^N$ be a set of size $N$ and let $X_n$ be a subset of size $n < N$ sampled uniformly at random without replacement from $\mathcal{X}$. Then, for $0 < \varepsilon < W_q(\mathcal{X})$ and $m = \min(n, N-n)$,*

$$\mathbb{P}\left[|W_q(X_n) - W_q(\mathcal{X})| \geq \varepsilon\right] \leq 2\exp\left(-\frac{2n\varepsilon^2}{(1-\frac{n}{N})(1+\frac{1}{m})}\Gamma(\varepsilon, a, b, q)\right), \tag{8}$$

*where*

$$\Gamma(\varepsilon, a, b, q) = \begin{cases} \frac{(1-2^q)^2 b^{2q-2}}{(a^q - b^q)^2} & q < 0 \\ \frac{1}{(b+\varepsilon)^2(\log b - \log a)^2} & q = 0 \\ \frac{q^2 a^{2q}}{(b+(1-q)\varepsilon)^2(b^q - a^q)^2} & 0 < q < 1 \\ \frac{1}{(b-a)^2} & q = 1 \\ \frac{q^2 a^{2q}}{(b+q\varepsilon)^2(b^q - a^q)^2} & q > 1 \end{cases} \tag{9}$$

*Similarly, for $q \in \{\pm\infty\}$:*

$$\mathbb{P}\left[|W_q(X_n) - W_q(\mathcal{X})| \geq \varepsilon\right] \leq 1 - \frac{n}{N}.$$

See Appendix A.2.1 for the full proof. Note that, for $q = 1$ (utilitarian rule), it is sufficient to have $n = N\left(1 + \frac{N}{2K}\right)^{-1}$ in Theorem 3. However, for $q \neq 1$, $\Gamma$ depends highly on $a$ and $b$ (respectively $U_{min}$ and $U_{max}$ in our setting). Worse, for $q = \pm\infty$, the bound depends linearly on $n$. Indeed, $q = -\infty$ corresponds to the egalitarian rule, which defines social welfare as the lowest welfare among individuals. As this individual might be unique, the probability of not selecting it in $I_n$ can be as high as $1 - \frac{n}{N}$. The same argument can be made for $q = +\infty$, which is why we purposefully avoid these scenarios in Theorems 2, 3 and 8. Regarding the error induced by the approximate model $\hat{p}$, we bound it using the following lemma:

**Lemma 6.** *Let $f : \mathcal{S} \to [f_{min}, f_{max}]$ be a bounded function with $0 \leq f_{min} \leq f_{max} < \infty$, and $p, \hat{p} \in \mathcal{P}(\mathcal{S})$ be two distributions such that $D_{KL}(p\|\hat{p}) \leq d \in \mathbb{R}$ and . Then*

$$\left|\mathbb{E}_{s\sim p}[f(s)] - \mathbb{E}_{s\sim\hat{p}}[f(s)]\right| \leq 2(f_{max} - f_{min})\sqrt{\min\{\frac{d}{2}, 1 - e^{-d}\}}.$$

See Appendix A.2.1 for the full proof. Combining Lemmas 5 and 6 along with other classical concentration inequalities, we can bound the error $|Q^*(s,a) - \hat{Q}^h(s,a)|$ with high probability. The last step of the proof is to quantify how this error affects the state value function $V^{\pi_{PAA}}$ (consequently $\mathcal{W}^{\pi_{PAA}}$ from Lemma 1), which can be done using the following results:

**Lemma 7.** *Let $\hat{Q}$ be a (randomized) approximation of $Q^*$ such that $|Q^*(s,a) - \hat{Q}(s,a)| \leq \varepsilon$ with probability at least $1 - \delta$ for any state-action pair $(s,a)$, with $\varepsilon > 0$ and $0 \leq \delta < 1$. Let $\pi_{\hat{Q}}$ be the greedy policy defined by $\pi_{\hat{Q}}(s) = \arg\max_{a\in\mathcal{A}}\hat{Q}(s,a)$. Then, for all states $s$:*

1) $V^*(s) - V^{\pi_{\hat{Q}}}(s) \leq \frac{2\varepsilon}{1-\gamma} + \gamma^k(V_{max} - V_{min})$    *with probability at least $1 - 2k\delta, \forall k \in \mathbb{N}^*$,*

2) $V^*(s) - V^{\pi_{\hat{Q}}}(s) \leq \frac{2\varepsilon + 2\delta(V_{max} - V_{min})}{1-\gamma}$                            *almost surely.*

These are fairly general results as they do not depend on how $\hat{Q}$ is derived. Statements closely related to 2) have already been shown [19, 35]. We provide a proof in Appendix A.2.1 for completeness.

## 3.3 Safe policies

Although Theorem 2 may initially inspire optimism regarding the title of the paper, the policy $\pi_{PAA}$ proposed in Eq. (7) is expensive for small $\varepsilon$, both in terms of sample complexity and in terms of required accuracy of the world model. A more efficient PAA policy derived in future work might partially solve the sample complexity issue, but the challenge of building predictive models of high accuracy remains untouched. In most realistic settings, $D_{KL}(p\|\hat{p})$ is imposed by the state-of-the-art knowledge upon which $\hat{p}$ is built, which implicitly restricts the achievable tolerance $\varepsilon$. Therefore, it seems unlikely that such policies could be used as a primary tool for social decisions, as their sole objective would be to maximize a dubious approximation of social welfare. On the other hand, even for large $\varepsilon$, we will show that we can use our PAA policy to adapt *any* black-box policy $\pi$ (e.g., a policy built on top of a LLM) into a *safe* policy, which we formally define as follows:

**Definition** (*Safe Policy*). Given $\omega \in [\mathcal{W}_{min}, \mathcal{W}_{max}]$ and $0 < \delta < 1$, a policy $\pi$ is $\delta$-$\omega$-safe if, for any current state $s$, the inequality $\mathbb{E}_{s' \sim p(\cdot|s,a)} \left[ \sup_{\pi'} \mathcal{W}^{\pi'}(s') \right] \geq \omega$ holds with probability at least $1 - \delta$ for any action $a$ such that $\pi(a|s) > 0$.

Intuitively, a safe policy ensures (with high probability and in expectation over the environment dynamics) that the society is not led in a destructive state, that is, a state which might generate high immediate satisfaction but where no policy can generate an expected future discounted social welfare of at least $\omega$. This is considerably weaker than the PAA requirements, as we are no longer concerned about social welfare optimality. The ability to adapt any black-box policy into a safe policy would allow to leverage their strengths while fully removing their brittleness (by bounding the probability of a destructive decision by any desired value $\delta > 0$). To this end, we use another type of policy:

**Definition** (*Restricted Version of a Black-Box Policy*). Let $\pi : \mathcal{S} \to \mathcal{A}$ be any policy and $\bar{\mathcal{A}}(s) \subseteq \mathcal{A}$ be restricted subsets of actions for all states $s$, with $\Pi(s) \triangleq \sum_{a \in \bar{\mathcal{A}}(s)} \pi(a|s)$. The restricted version $\bar{\pi}$ of $\pi$ is defined as

$$\bar{\pi}(a|s) \triangleq \begin{cases} 0 & a \in \mathcal{A} \setminus \bar{\mathcal{A}}(s) \quad \text{or} \quad \Pi(s) = 0, \\ \frac{\pi(a|s)}{1-\Pi(s)} & a \in \bar{\mathcal{A}}(s) \quad \text{and} \quad 0 < \Pi(s) < 1, \\ \pi(a|s) & a \in \bar{\mathcal{A}}(s) \quad \text{and} \quad \Pi(s) = 1, \end{cases} \tag{10}$$

This is similar to *action masking* presented in [20]. It might happen that $\bar{\pi}(a|s) = 0$ for all actions $a$, in which case it stops operating. However, if this happens, we have the guarantee that, with high probability, the society is currently not in a destructive state. The challenge lies in finding what are the subset of safe actions for every $s$. Our proposed method to safeguard any policy is the following:

**Theorem 8** (Safeguarding a Black-Box Policy). *Given a SMDP $\mathcal{M}_{\mathcal{I}} = (\mathcal{S}, \mathcal{A}, p, W_q, \mathbf{u}, \gamma)$ with $q \in \mathbb{R}$, a predictive model $\hat{p}$ and desired tolerances $\omega \in [\mathcal{W}_{min}, \mathcal{W}_{max}]$ and $0 < \delta < 1$, define $\hat{Q}_\omega(s,a) \triangleq \hat{Q}^H(s,a; K, C, I_n)$ with $\hat{Q}^H$ given in Eq. (6) and any $H, K, C, n \geq 1$. For any policy $\pi$, let $\pi_{safe}$ be the restricted version of $\pi$ obtained with the restricted subsets $\mathcal{A}_{safe}(s) \triangleq \{a \in \mathcal{A} : \hat{Q}_\omega(s,a) \geq \gamma\omega + U_{max} + \alpha\}$, where*

$$\alpha \triangleq \frac{2\Delta U d'}{(1-\gamma)^2} + \frac{\sqrt{\ln\left(\frac{12(C|\mathcal{A}|)^{H-1}}{\delta}\right)}}{1-\gamma} \left( \sqrt{\frac{N-n}{nN\Gamma_{max}}} + \sqrt{\frac{\Delta U^2}{2K}} + \gamma\sqrt{\frac{\Delta U^2}{2C(1-\gamma)^2}} \right) + \frac{\gamma^H \Delta U}{1-\gamma},$$

*and with the shortened notation $\Gamma_{max} \triangleq \Gamma(U_{max}, U_{min}, U_{max}, q)$, $d' \triangleq \sqrt{\min\{\frac{d}{2}, 1 - e^{-d}\}}$ and $d \triangleq \sup_{(s,a)} D_{KL}(p(\cdot|s,a)\|\hat{p}(\cdot|s,a))$. Then $\pi_{safe}$ is $\delta$-$\omega$-safe.*

See Appendix A.2.3 for the full proof. While we are, in theory, not restricted by the statistical accuracy of the world model to find a safe version of any black-box policy, low statistical accuracy will, in practice, drastically reduce the number of verifiably safe actions, which at some point will render the safe policy obsolete (as it will refuse to take any action).

# 4 Related work

**MDP for social choice**   MDPs have already been used in the context of social decision processes. For instance, Parkes and Procaccia [28] (and more recently [21]) use *social choice MDPs* in order to tackle decision-making under dynamic preferences. However, their setting is significantly different from ours: In their work, the state space $\mathcal{S}$ is the set of preference profiles $\mathcal{U}^N$, and $p$ dictate how these preferences evolve based on the outcome selected by the social choice functional (the policy). Another relevant line of study is that of preference-based reinforcement learning (PbRL) [40], where the traditional numerical reward of the underlying MDP is replaced with relative preferences between state-action trajectories. While social decision processes can also be cast as a PbRL problem, no previous work has attempted to formally and quantitatively define social alignment in that setting.

**AI safety**   A long line of work has attempted to address the challenge of building verifiably safe autonomous systems [2, 30]. In the context of MDPs, safe reinforcement learning (RL) tackles this challenge by introducing safety constraints [1]. See [15, 20] for comprehensive surveys on the topic. However, RL relies on exploration, which is not allowed in our setting. On the other hand, existing planning methods (where exploration is not needed if a world model $\hat{p}$ is available) do not relate the accuracy of $\hat{p}$ to the validity of the desired safety guarantees, as they mostly assume that $p$ is known.

**Alignment**   The goal of alignment can be entirely different based on the context [13]. Recent research on this topic has primarily focused on aligning large language models (LLMs) [17] using human feedback [6, 26] and derivatives [11, 39, 5]. While some work has attempted to tackle LLM alignment from a social choice perspective [22], the issue of aligning the meaning of generated text is, by nature, both qualitative and subjective, and therefore separate from ours. A setting closer to our work is the *value alignment problem* [34, 23], based on the theory of basic human values [31], where the preferences of individuals (over social states) are assumed to be guided by a predefined set of common values, and where the goal is to find "norms" (i.e., hard-coded logical constraints on actions) that guide society towards states that maximize these values. The alignment of these norm can be quantified by measuring the level of these values in the subsequent states, and the alignment of an autonomous system is simply given by the alignment of the norms it follows. However, this measure is intractable in most realistic settings, as it must be computed over all possible state trajectories [34].

# 5 Limitations and future work

From a practical standpoint, the main challenge lies in building a reliable world model $\hat{p}$, since PAA guarantees depend on its statistical accuracy, which can only be measured exactly if the true world model $p$ is known. In practice, a conservative estimate of this accuracy could be used instead. Another limitation arises from the dependence on the informational basis of utilities, a philosophical question that falls outside the scope of this paper but that is common to all systems involving human feedback. A third practical limitation is the assumptions that individuals can observe and evaluate the entire social states when reporting their utilities. Future work could extend our analysis to the setting of partially observable MDP (POMDP) [42], for instance. Finally, the assumption that individuals have static preferences can also be challenged, but it is not clear how evolving preferences can be modeled, let alone factored in our analysis. From a theoretical perspective, the complexity results presented in the various theorems are poor for $q \neq 1$ and $\gamma \approx 1$. These dependencies are hard to improve, as they relate to a known property of the power mean [7] and the ability to foresee the future, respectively. Lastly, while we make no assumption about the distribution of utilities $\mathbf{u}$, one could investigate how such assumptions might improve these complexities (e.g., using Bernstein-Serfling inequality [4]).

# 6 Conclusion

We present a formal and quantitative definition of alignment in the context of social choice, leading to the concept of *probably approximately aligned* policies. Using an approximate world model, we derive sufficient conditions for such policies to exist (and be *computable*). In addition, we introduce the relaxed concept of *safe* policies, and we present a method to make any policy safe by restricting its action space. Overall, this work provides a first attempt at rigorously defining the alignment of governing autonomous agents, and at quantifying the resources ($n$, $K$, $C$, $H$) and knowledge ($\hat{p}$) needed to enforce the desired level of alignment ($\varepsilon$) or safety ($\omega$) with high confidence ($\delta$).

## Acknowledgments and Disclosure of Funding

Special thanks to Prof. Dr. Hans Gersbach for his insightful and thorough feedback, which contributed to the refinement and clarity of this paper.

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

# A  Appendix

## A.1  Utility and Social Choice Theory

### A.1.1  Measuring and comparing utilities

Measurability of $u \in \mathcal{U}$ can either be *cardinal*, where one can numerically measure absolute levels of satisfaction for a given state up to positive affine transformations, or *ordinal*, where one can only hope to rank states, meaning one can numerically measure utilities only up to increasing monotone transformations [16, 36, 27]. Additional dichotomization becomes possible when comparability is taken into account [29, 9, 32]. In a nutshell, utilities are either *ordinal non-comparable* (ONC), where individuals transform their intrinsic utilities differently when reporting; *cardinal non-comparable* (CNC), which is similar to ONC but with cardinal measurability), *ordinal level comparable* (OLC), where individuals transform their intrinsic utilities similarly when reporting; *cardinal unit comparable* (CUC), in which the affine transforms of the individuals have identical scaling factor but different bias; *cardinal fully comparable* (CFC), where individuals have the same affine transform; or *cardinal ratio-scale* comparable (CRS), in which all transforms are unbiased with the same scaling factor. Note that CRS is stronger than CFC in the sense that one can have statements such has "Bob is $x$ times more satisfied than Alice" under the CRS assumption. In this paper, we refer to this classification as the *informational basis* of utilities.

### A.1.2  SWFL properties and implications

Here is a list of the most common properties imposed on a given SWFL $f : \mathcal{D}_f \to \mathcal{R}_\mathcal{S}$, along with their definitions. We denote $\preceq_{f(\mathbf{u})}$ the binary preference relationship corresponding to $f(\mathbf{u})$, that is, $s \preceq_{f(\mathbf{u})} s'$ if and only if $s'$ ranks equally or higher than $s$ in $f(\mathbf{u})$.

- (U) Universality or unrestricted domain: $\mathcal{D}_f = \mathcal{U}^N$.

- (IIA) Independence of Irrelevant Alternatives: For every $\mathbf{u}, \mathbf{u}' \in \mathcal{U}^N$ and $\mathcal{S}' \subseteq \mathcal{S}$, if $\mathbf{u}(s) = \mathbf{u}'(s)$ for all $s \in \mathcal{S}'$, then $f_{\mathcal{S}'}(\mathbf{u}) = f_{\mathcal{S}'}(\mathbf{u}')$, where $f_{\mathcal{S}'}(\mathbf{u})$ is the partial ranking obtained after excluding $\mathcal{S} \setminus \mathcal{S}'$ from $f(\mathbf{u})$.

- (WP) Weak Pareto criterion or unanimity: For all pairs $s, s' \in \mathcal{S}$, if $\mathbf{u}(s) > \mathbf{u}(s')$, then $s$ ranks strictly higher than $s'$ in $f(\mathbf{u})$.

- (WPI) Weak Pareto with Indifference criterion: For all pairs $s, s' \in \mathcal{S}$, if $\mathbf{u}(s) \geq \mathbf{u}(s')$, then $s$ ranks equally or higher than $s'$ in $f(\mathbf{u})$.

- (SP) Strong Pareto criterion: For all pairs $s, s' \in \mathcal{S}$, if there exists $i \in \mathcal{I}$ such that $u_i(s) > u_i(s')$ and $u_j(s) \geq u_j(s'), \forall j \in \mathcal{I} \setminus \{i\}$, then $s$ ranks strictly higher than $s'$ in $f(\mathbf{u})$.

- (NI) Non-Imposition: For all $R \in \mathcal{R}$, there exists $\mathbf{u} \in \mathcal{D}_f$ such that $f(\mathbf{u}) = R$.

- (C) Continuity: For any $\mathbf{v} \in \mathbb{R}^N$ and $s, s' \in \mathcal{S}$, the sets

$$\{\mathbf{v}' \in \mathbb{R}^N : \mathbf{u}(s') = \mathbf{v}', \mathbf{u}(s) = \mathbf{v} \text{ and } s \preceq_{f(\mathbf{u})} s' \text{ for some } \mathbf{u} \in \mathcal{U}^N\},$$

  and

$$\{\mathbf{v}' \in \mathbb{R}^N : \mathbf{u}(s') = \mathbf{v}', \mathbf{u}(s) = \mathbf{v} \text{ and } s' \preceq_{f(\mathbf{u})} s \text{ for some } \mathbf{u} \in \mathcal{U}^N\}$$

  are closed.

- (WC) Weak Continuity: For any $\mathbf{u} \in \mathcal{U}^N$ and $\varepsilon \in \mathbb{R}_+^N$, there exists $\mathbf{u}' \in \mathcal{U}^N$ such that $f(\mathbf{u}) = f(\mathbf{u}')$ and $\mathbf{0} < \mathbf{u}(s) - \mathbf{u}'(s) < \varepsilon$ for all $s \in \mathcal{S}$ (component-wise inequalities).

- (N) Neutrality or welfarism: For all quadruples $s, s', t, t' \in \mathcal{S}$, if $\mathbf{u}, \mathbf{u}' \in \mathcal{U}^N$ are such that $\mathbf{u}(s) = \mathbf{u}'(s')$ and $\mathbf{u}(t) = \mathbf{u}'(t')$, then $f(\mathbf{u})$ and $f(\mathbf{u}')$ agree on the partial rankings of $(s, t)$ and $(s', t')$ (i.e., either $s$ and $s'$ are preferred, or either $t$ and $t'$).

- (A) Anonymity or symmetry: For all $\mathbf{u} \in \mathcal{U}^N$, $f(\mathbf{u}) = f(\mathbf{u}')$ with $\mathbf{u}'$ any permutation of $\mathbf{u}$.

- (ND) Non-Dictatorship: There is no single individual $i$ such that $\preceq_i = \preceq_{f(\mathbf{u})}$ for any $\mathbf{u}$.

- (IC) Incentive compatibility: It is in the best interest of each individual to report their true preferences (i.e., there is no tactical voting).

(S) Separability or independence of unconcerned agents: For all $\mathbf{u}, \mathbf{u}' \in \mathcal{U}$, if there exists $\mathcal{I}' \subseteq \mathcal{I}$ such that $u_i = \alpha \in \mathbb{R}$ and $u_i' = \alpha' \in \mathbb{R}$ for $i \in \mathcal{I}'$, and $u_i = u_i'$ for $i \in \mathcal{I} \setminus \mathcal{I}'$, then $f(\mathbf{u}) = f(\mathbf{u}')$. $\mathcal{I}'$ is the set of unconcerned agents.

(PDT) Pigou-Dalton Transfer principle: Define $\bar{u}(s) = \sum_{i \in \mathcal{I}} u_i(s)$. For all $\mathbf{u} \in \mathcal{U}^N$ and $s, t \in \mathcal{S}$ such that $\bar{u}(s) = \bar{u}(t)$, if $|u_i(s) - \bar{u}(s)| \leq |u_i(t) - \bar{u}(t)|$ for all $i \in \mathcal{I}$, then $s$ ranks equally or higher than $t$ in $f(\mathbf{u})$.

(XI) Informational basis Invariance (XI): The SWFL is invariant under the given measurability and comparability assumptions: (ONCI), (CNCI), (CUCI), (CFCI), (CRSI), (OLCI). That is, $f(\mathbf{u}) = f(\mathbf{u}')$ for any two utility profile $\mathbf{u}, \mathbf{u}'$ that are undistinguishable under the given informational basis.

We now provide a short, intuitive explanation for the properties (U), (XI), (IIA), (WP) and (A) that we impose in this work. Enforcing unrestricted domains ensures that the SWFL always outputs a ranking. Informational basis invariance ensures that the SWFL outputs the same ranking for two preference profiles that are indistinguishable under the given measurability and comparability assumptions. Independence of irrelevant alternatives ensures that the rankings are robust, in the sense that they are not incoherently affected by removing or adding other options. The weak Pareto criterion ensures that the SWFL represents reasonably well the preference of society (if an outcome is unanimously preferred over another, then it must also be preferred at the social level). Anonymity ensures that all individuals have equal influence on the social ranking. Lastly, we recall a few important results (assuming $|\mathcal{S}| \geq 3$).

- (CRSI) $\Rightarrow$ (CFCI) $\Rightarrow$ (CUCI) $\Rightarrow$ (CNCI) $\Rightarrow$ (ONCI). Additionally (OLCI) $\Rightarrow$ (ONCI).

- (XI) + (U) + (IIA) + (WP) $\Rightarrow$ There exists $W : \mathbb{R}^N \to \mathbb{R}$ such that $W(\mathbf{u}(s)) > W(\mathbf{u}(s'))$ implies that $s$ ranks strictly higher than $s'$ in $f(\mathbf{u})$. This is important as it states that the best social state $s$ must maximize $W$. In other words, even if $f$ is not neutral, the non-welfarism characteristics are of a secondary importance and can only break ties between $s$ and $t$ such that $W(\mathbf{u}(s)) = W(\mathbf{u}(t))$. Maximization can be made sufficient if one imposes (W) on $f$ (e.g., by replacing (WP) by (WPI) or more drastically by imposing continuity on $f$).

- Arrow's impossibility theorem [3]: (ONCI) + (U) + (WP) + (I) $\Rightarrow$ ¬(ND). While this theorem seems to prevent any hope of finding good SWFLs, its statement is strongly dependent on the (often hidden) ONCI assumptions. Indeed, it is challenging to find a good social aggregator that only knows rankings of alternatives. Strengthening the measurability and comparability assumptions (and thus narrowing the informational basis invariance property) allows to find SWFLs that are non-dictatorial.

- (A) $\Rightarrow$ (ND).

- Gibbard–Satterthwaite theorem (single winner elections): (ONCI) + (IC) $\Rightarrow$ ¬(ND).

- (SP) $\Rightarrow$ (WP) but the converse is not true.

- (XI) $\Rightarrow$ (WC)

### A.1.3 Power mean and SWFL correspondence

Table 1: The different social welfare functions (SWF) corresponding to a SWFL that satisfies (U), (IIA), (WP), (A) and (XI) for the different informational bases [29].

| (XI) | SWF ($W$) | Power mean |
|---|---|---|
| (ONCI) or (CNCI) | Impossibility (Arrow [3]) | - |
| (CUCI) | $\sum_{i \in \mathcal{I}} u_i(s)$ | $q = 1$ |
| (OLCI) | $\min_{i \in \mathcal{I}} u_i(s)$ or $\max_{i \in \mathcal{I}} u_i(s)$ | $q \in \{\pm\infty\}$ |
| (CFCI) | $\min_{i \in \mathcal{I}} u_i(s)$, $\max_{i \in \mathcal{I}} u_i(s)$ or $\sum_{i \in \mathcal{I}} u_i(s)$ | $q \in \{\pm\infty, 1\}$ |
| (CRSI) | $\min_{i \in \mathcal{I}} u_i(s)$, $\max_{i \in \mathcal{I}} u_i(s)$, $\sum_{i \in \mathcal{I}} u_i(s)^q$ or $\sum_{i \in \mathcal{I}} \log[u_i(s)]$ | $q \in \mathbb{R} \cup \{\pm\infty\}$ |

## A.2 Proofs

### A.2.1 Intermediate results

**Lemma 1.** *For any SMDP $\mathcal{M}_\mathcal{I} = (\mathcal{S}, \mathcal{A}, p, W_q, \mathbf{u}, \gamma)$, the expected future discounted social welfare of a policy $\pi$ is the state value function of $\pi$ in the MDP $\mathcal{M} = (\mathcal{S}, \mathcal{A}, p, r_\mathcal{I}, \gamma)$, with $r_\mathcal{I}$ set in Eq. (2).*

*Proof.* Let $\tau_t = s_0, a_0, s_1, a_1, ..., s_t$ denote a truncated trajectory up to time $t$. From the definitions of $V^\pi(s; \mathcal{M})$ and $r_\mathcal{I}$, we have

$$
\begin{aligned}
V^\pi(s; \mathcal{M}) &= \mathbb{E}_{\tau \sim p_\tau(\cdot|\pi, s_0 = s)} \left[ \sum_{t=0}^{\infty} \gamma^t r_\mathcal{I}(s_t, a_t) \right] \\
&= \mathbb{E}_{\tau \sim p_\tau(\cdot|\pi, s_0 = s)} \left[ \sum_{t=0}^{\infty} \gamma^t \mathbb{E}_{s' \sim p(\cdot|s_t, a_t)}[W_q(\mathbf{u}(s'))] \right] \\
&= \sum_{t=0}^{\infty} \gamma^t \mathbb{E}_{\tau \sim p_\tau(\cdot|\pi, s_0 = s)} \left[ \mathbb{E}_{s' \sim p(\cdot|s_t, a_t)}[W_q(\mathbf{u}(s'))] \right] \\
&= \sum_{t=0}^{\infty} \gamma^t \mathbb{E}_{\tau_t \sim p_{\tau_t}(\cdot|\pi, s_0 = s), a_t \sim \pi(\cdot|s_t)} \left[ \mathbb{E}_{s' \sim p(\cdot|s_t, a_t)}[W_q(\mathbf{u}(s'))] \right] \\
&= \sum_{t=0}^{\infty} \gamma^t \mathbb{E}_{\tau_{t+1} \sim p_{\tau_{t+1}}(\cdot|\pi, s_0 = s)} \left[ W_q(\mathbf{u}(s_{t+1})) \right] \\
&= \sum_{t=0}^{\infty} \gamma^t \mathbb{E}_{\tau \sim p_\tau(\cdot|\pi, s_0 = s)} \left[ W_q(\mathbf{u}(s_{t+1})) \right] \\
&= \mathbb{E}_{\tau \sim p_\tau(\cdot|\pi, s_0 = s)} \left[ \sum_{t=0}^{\infty} \gamma^t W_q(\mathbf{u}(s_{t+1})) \right] \\
&= \mathcal{W}^\pi(s; \mathcal{M}_\mathcal{I}).
\end{aligned}
$$

$\square$

**Lemma 5.** *Let $W_q$ be the power-mean defined in Eq. (1), with $q \in \mathbb{R}$. Given $a, b \in \mathbb{R}_+^*$ (or $\mathbb{R}_+$ for $q = 1$) such that $a < b$, let $\mathcal{X} \in [a, b]^N$ be a set of size $N$ and let $X_n$ be a subset of size $n < N$ sampled uniformly at random without replacement from $\mathcal{X}$. Then, for $0 < \varepsilon < W_q(\mathcal{X})$ and $m = \min(n, N - n)$,*

$$
\mathbb{P}\left[|W_q(X_n) - W_q(\mathcal{X})| \geq \varepsilon\right] \leq 2 \exp\left( -\frac{2n\varepsilon^2}{(1 - \frac{n}{N})(1 + \frac{1}{m})} \Gamma(\varepsilon, a, b, q) \right), \tag{8}
$$

*where*

$$
\Gamma(\varepsilon, a, b, q) = \begin{cases}
\frac{(1-2^q)^2 b^{2q-2}}{(a^q - b^q)^2} & q < 0 \\
\frac{1}{(b+\varepsilon)^2 (\log b - \log a)^2} & q = 0 \\
\frac{q^2 a^{2q}}{(b+(1-q)\varepsilon)^2 (b^q - a^q)^2} & 0 < q < 1 \\
\frac{1}{(b-a)^2} & q = 1 \\
\frac{q^2 a^{2q}}{(b+q\varepsilon)^2 (b^q - a^q)^2} & q > 1
\end{cases} \tag{9}
$$

*Similarly, for $q \in \{\pm\infty\}$:*

$$
\mathbb{P}\left[|W_q(X_n) - W_q(\mathcal{X})| \geq \varepsilon\right] \leq 1 - \frac{n}{N}.
$$

*Proof.* To simplify the notation, we write $S = W_q(X_n)$ and $\mu = W_q(\mathcal{X})$ such that

$$
\begin{aligned}
S^q &= \frac{1}{n} \sum_{x_i \in X_n} x_i^q, & \mu^q &= \frac{1}{N} \sum_{x_i \in \mathcal{X}} x_i^q && \text{for } q \in \mathbb{R}^* \\
\log S &= \frac{1}{n} \sum_{x_i \in X_n} \log x_i^q & \log \mu &= \frac{1}{N} \sum_{x_i \in \mathcal{X}} \log x_i^q && \text{for } q = 0
\end{aligned}
$$

For $q \in \mathbb{R}$, we have

$$\mathbb{P}\left[|S - \mu| \geq \varepsilon\right] = 1 - \mathbb{P}\left[\mu - \varepsilon < S < \mu + \varepsilon\right]$$
$$= \begin{cases} 1 - \mathbb{P}\left[(\mu + \varepsilon)^q < S^q < (\mu - \varepsilon)^q\right] & q < 0 \\ 1 - \mathbb{P}\left[\log(\mu - \varepsilon) < \log S < \log(\mu + \varepsilon)\right] & q = 0 \\ 1 - \mathbb{P}\left[(\mu - \varepsilon)^q < S^q < (\mu + \varepsilon)^q\right] & q > 0 \end{cases} \tag{11}$$

Therefore:

- For $q < 0$:

$$\mathbb{P}\left[(\mu + \varepsilon)^q < S^q < (\mu - \varepsilon)^q\right] = \mathbb{P}\left[\left(1 + \frac{\varepsilon}{\mu}\right)^q \mu^q < S^q < \left(1 - \frac{\varepsilon}{\mu}\right)^q \mu^q\right]$$
$$\geq \mathbb{P}\left[\left(1 - \frac{(1 - 2^q)\varepsilon}{\mu}\right)\mu^q < S^q < \left(1 + \frac{(1 - 2^q)\varepsilon}{\mu}\right)\mu^q\right]$$
$$= \mathbb{P}\left[\mu^q - (1 - 2^q)\mu^{q-1}\varepsilon < S^q < \mu^q + (1 - 2^q)\mu^{q-1}\varepsilon\right]$$
$$\geq \mathbb{P}\left[\mu^q - (1 - 2^q)b^{q-1}\varepsilon < S^q < \mu^q + (1 - 2^q)b^{q-1}\varepsilon\right]$$
$$= \mathbb{P}\left[|S^q - \mu^q| < (1 - 2^q)b^{q-1}\varepsilon\right] \tag{12}$$

where we have used the following approximation (holding for $0 < x \leq 1$ and $q < 0$):

$$(1 + x)^q \leq 1 - (1 - 2^q)x < 1 < 1 + (1 - 2^q)x \leq (1 - x)^q.$$

Combining Eq. (11) and (12), and using the Hoeffding-Serfling inequality (Lemma 9) after observing that $b^q \leq x_i^q \leq a^q$ for all $i$, we get

$$\mathbb{P}\left[|S - \mu| \geq \varepsilon\right] \leq \mathbb{P}\left[|S^q - \mu^q| \geq (1 - 2^q)b^{q-1}\varepsilon\right]$$
$$\leq 2\exp\left(-\frac{2n(1 - 2^q)^2 b^{2q-2}\varepsilon^2}{(1 - \frac{n}{N})(1 + \frac{1}{m})(a^q - b^q)^2}\right).$$

- For $q = 0$:

$$\mathbb{P}\left[\log(\mu - \varepsilon) < \log S < \log(\mu + \varepsilon)\right] = \mathbb{P}\left[\log(1 - \frac{\varepsilon}{\mu}) < \log S - \log \mu < \log(1 + \frac{\varepsilon}{\mu})\right]$$
$$\geq \mathbb{P}\left[-\frac{\varepsilon}{\mu + \varepsilon} < \log S - \log \mu < \frac{\varepsilon}{\mu + \varepsilon}\right]$$
$$\geq \mathbb{P}\left[-\frac{\varepsilon}{b + \varepsilon} < \log S - \log \mu < \frac{\varepsilon}{b + \varepsilon}\right], \tag{13}$$

where we have used the following approximation (holding for $0 < x < 1$):

$$\log(1 - x) \leq -\frac{x}{1 + x} < 0 < \frac{x}{1 + x} \leq \log(1 + x).$$

Combining Eq. (11) and (13) and using the Hoeffding-Serfling inequality (Lemma 9) after observing that $\log a \leq \log x_i^q \leq \log b$ for all $i$, we get

$$\mathbb{P}\left[|S - \mu| \geq \varepsilon\right] \leq \mathbb{P}\left[|\log S - \log \mu| \geq \frac{\varepsilon}{b + \varepsilon}\right]$$
$$\leq 2\exp\left(-\frac{2n\varepsilon^2}{(1 - \frac{n}{N})(1 + \frac{1}{m})(b + \varepsilon)^2(\log b - \log a)^2}\right).$$

- For $0 < q < 1$:

$$\mathbb{P}\left[(\mu - \varepsilon)^q < S^q < (\mu + \varepsilon)^q\right] = \mathbb{P}\left[\left(1 - \frac{\varepsilon}{\mu}\right)^q \mu^q < S^q < \left(1 + \frac{\varepsilon}{\mu}\right)^q \mu^q\right]$$

$$\geq \mathbb{P}\left[-\frac{q\varepsilon\mu^q}{\mu + (1-q)\varepsilon} < S^q - \mu^q < \frac{q\varepsilon\mu^q}{\mu + (1-q)\varepsilon}\right]$$

$$\geq \mathbb{P}\left[-\frac{q\varepsilon\mu^q}{b + (1-q)\varepsilon} < S^q - \mu^q < \frac{q\varepsilon\mu^q}{b + (1-q)\varepsilon}\right]$$

$$\geq \mathbb{P}\left[-\frac{qa^q\varepsilon}{b + (1-q)\varepsilon} < S^q - \mu^q < \frac{qa^q\varepsilon}{b + (1-q)\varepsilon}\right]$$

$$= \mathbb{P}\left[|S^q - \mu^q| < \frac{qa^q\varepsilon}{b + (1-q)\varepsilon}\right] \qquad (14)$$

where we have used the following approximation (holding for $0 < x \leq 1$ and $0 < q < 1$):

$$(1-x)^q \leq 1 - \frac{qx}{1+(1-q)x} < 1 < 1 + \frac{qx}{1+(1-q)x} \leq (1+x)^q.$$

Combining Eq. (11) and (14), and using the Hoeffding-Serfling inequality (Lemma 9) after observing that $a^q \leq x_i^q \leq b^q$ for all $i$, we get

$$\mathbb{P}\left[|S - \mu| \geq \varepsilon\right] \leq \mathbb{P}\left[|S^q - \mu^q| \geq \frac{qa^q\varepsilon}{b + (1-q)\varepsilon}\right]$$

$$\leq 2\exp\left(-\frac{2nq^2a^{2q}\varepsilon^2}{(1 - \frac{n}{N})(1 + \frac{1}{m})(b + (1-q)\varepsilon)^2(b^q - a^q)^2}\right).$$

- For $q = 1$: We directly apply the Hoeffding-Serfling inequality (Lemma 9) to obtain

$$\mathbb{P}\left[|S - \mu| \geq \varepsilon\right] \leq 2\exp\left(-\frac{2n\varepsilon^2}{(1 - \frac{n}{N})(1 + \frac{1}{m})(b - a)}\right).$$

- For $q > 1$:

$$\mathbb{P}\left[(\mu - \varepsilon)^q < S^q < (\mu + \varepsilon)^q\right] = \mathbb{P}\left[\left(1 - \frac{\varepsilon}{\mu}\right)^q \mu^q < S^q < \left(1 + \frac{\varepsilon}{\mu}\right)^q \mu^q\right]$$

$$\geq \mathbb{P}\left[\left(1 - \frac{q\varepsilon}{\mu + q\varepsilon}\right)\mu^q < S^q < \left(1 + \frac{q\varepsilon}{\mu + q\varepsilon}\right)\mu^q\right]$$

$$\geq \mathbb{P}\left[\left(1 - \frac{q\varepsilon}{b + q\varepsilon}\right)\mu^q < S^q < \left(1 + \frac{q\varepsilon}{b + q\varepsilon}\right)\mu^q\right]$$

$$\geq \mathbb{P}\left[\mu^q - \frac{qa^q\varepsilon}{b + q\varepsilon} < S^q < \mu^q + \frac{qa^q\varepsilon}{b + q\varepsilon}\right]$$

$$= \mathbb{P}\left[|S^q - \mu^q| < \frac{qa^q\varepsilon}{b + q\varepsilon}\right] \qquad (15)$$

where we have used the following approximation (holding for $0 < x \leq 1$ and $q > 1$):

$$(1-x)^q \leq 1 - \frac{qx}{1+qx} < 1 < 1 + \frac{qx}{1+qx} \leq (1+x)^q.$$

Combining Eq. (11) and (15), and using the Hoeffding-Serfling inequality (Lemma 9) after observing that $a^q \leq x_i^q \leq b^q$ for all $i$, we get

$$\mathbb{P}\left[|S - \mu| \geq \varepsilon\right] \leq \mathbb{P}\left[|S^q - \mu^q| \geq \frac{qa^q\varepsilon}{b + q\varepsilon}\right]$$

$$\leq 2\exp\left(-\frac{2nq^2a^{2q}\varepsilon^2}{(1 - \frac{n}{N})(1 + \frac{1}{m})(b + q\varepsilon)^2(b^q - a^q)^2}\right).$$

- For $q = +\infty$: Since we make no assumptions on the distributions of $x_i$ other than $a \leq x_i \leq b$, we can only guarantee that $|\max_{x_i \in X_n} x_i - \max_{x_i \in \mathcal{X}} x_i| < \varepsilon$ if $x_{max} = \arg\max_{x_i \in \mathcal{X}} x_i$ is sampled in $X_n$. This happens with probability at least $\frac{n}{N}$ (the maximum might not be unique), and thus $\mathbb{P}\left[|\max_{x_i \in X_n} x_i - \max_{x_i \in \mathcal{X}} x_i| \geq \varepsilon\right] \leq 1 - \frac{n}{N}$.

- For $q = -\infty$: Same analysis as for $q = +\infty$.

$\square$

**Lemma 6.** *Let* $f : \mathcal{S} \to [f_{min}, f_{max}]$ *be a bounded function with* $0 \leq f_{min} \leq f_{max} < \infty$, *and* $p, \hat{p} \in \mathcal{P}(\mathcal{S})$ *be two distributions such that* $D_{KL}(p\|\hat{p}) \leq d \in \mathbb{R}$ *and . Then*

$$\left|\mathbb{E}_{s\sim p}[f(s)] - \mathbb{E}_{s\sim\hat{p}}[f(s)]\right| \leq 2(f_{max} - f_{min})\sqrt{\min\{\frac{d}{2}, 1 - e^{-d}\}}.$$

*Proof.*

$$\begin{aligned}
\left|\mathbb{E}_{s\sim p}[f(s)] - \mathbb{E}_{s\sim\hat{p}}[f(s)]\right| &= \left|\int_{\mathcal{S}} f(s)p(s)\,\mathrm{d}s - \int_{\mathcal{S}} f(s)\hat{p}(s)\,\mathrm{d}s\right| \\
&= \left|\int_{\mathcal{S}} (f(s) - f_{min})p(s)\,\mathrm{d}s - \int_{\mathcal{S}} (f(s) - f_{min})\hat{p}(s)\,\mathrm{d}s\right| \\
&= \left|\int_{\mathcal{S}} (f(s) - f_{min})(p(s) - \hat{p}(s))\,\mathrm{d}s\right| \\
&\leq \int_{\mathcal{S}} |f(s) - f_{min}||p(s) - \hat{p}(s)|\,\mathrm{d}s \\
&\leq (f_{max} - f_{min})\int_{\mathcal{S}} |p(s) - \hat{p}(s)|\,\mathrm{d}s \\
&= 2(f_{max} - f_{min})\delta(p, \hat{p})
\end{aligned}$$

where we have used the definition of the total variation distance: $\delta(p, \hat{p}) = \frac{1}{2}\int_{\mathcal{S}} |p(s) - \hat{p}(s)|\,\mathrm{d}s$. By Pinsker's inequality, we have

$$\delta(p, \hat{p}) \leq \sqrt{\frac{1}{2}D_{KL}(p\|\hat{p})}.$$

Additionally, by Bretagnolle and Huber's inequality:

$$\delta(p, \hat{p}) \leq \sqrt{1 - e^{-D_{KL}(p\|\hat{p})}}.$$

The result follows from the assumption $D_{KL}(p\|\hat{p}) \leq d$. $\square$

**Lemma 7.** *Let* $\hat{Q}$ *be a (randomized) approximation of* $Q^*$ *such that* $|Q^*(s, a) - \hat{Q}(s, a)| \leq \varepsilon$ *with probability at least* $1 - \delta$ *for any state-action pair* $(s, a)$, *with* $\varepsilon > 0$ *and* $0 \leq \delta < 1$. *Let* $\pi_{\hat{Q}}$ *be the greedy policy defined by* $\pi_{\hat{Q}}(s) = \arg\max_{a \in \mathcal{A}} \hat{Q}(s, a)$. *Then, for all states* $s$:

1) $V^*(s) - V^{\pi_{\hat{Q}}}(s) \leq \dfrac{2\varepsilon}{1 - \gamma} + \gamma^k(V_{max} - V_{min})$ *with probability at least* $1 - 2k\delta, \forall k \in \mathbb{N}^*$,

2) $V^*(s) - V^{\pi_{\hat{Q}}}(s) \leq \dfrac{2\varepsilon + 2\delta(V_{max} - V_{min})}{1 - \gamma}$ *almost surely.*

*Proof.* First, note that if $|Q^*(s, a) - \hat{Q}(s, a)| \leq \varepsilon$ with probability at least $1 - \delta$ for all state-action pairs $(s, a)$, then $Q^*(s, \pi^*(s)) - Q^*(s, \pi_{\hat{Q}}(s)) \leq 2\varepsilon$ with probability at least $1 - 2\delta$ since

$$Q^*(s, \pi^*(s)) \leq \hat{Q}(s, \pi^*(s)) + \varepsilon \leq \hat{Q}(s, \pi_{\hat{Q}}(s)) + \varepsilon \leq [Q^*(s, \pi_{\hat{Q}}(s)) + \varepsilon] + \varepsilon.$$

The factor 2 in the probability comes from the fact that we need the $\hat{Q}$ estimates to be accurate for both actions $\pi^*(s)$ and $\pi_{\hat{Q}}(s)$. For the first inequality, note that, with probability at least $1 - 2k\delta$, the

first $k$ estimates of $\hat{Q}$ are $2\varepsilon$-accurate for actions $\pi^*(s_t)$ and $\pi_{\hat{Q}}(s_t)$, $t = 0, ..., k-1$. If this happens, then we have

$$
\begin{aligned}
V^*(s) - V^{\pi_{\hat{Q}}}(s) &= Q^*(s, \pi^*(s)) - Q^{\pi_{\hat{Q}}}(s, \pi_{\hat{Q}}(s)) \\
&\leq 2\varepsilon + Q^*(s, \pi_{\hat{Q}}(s)) - Q^{\pi_{\hat{Q}}}(s, \pi_{\hat{Q}}(s)) \\
&= 2\varepsilon + \gamma \mathbb{E}_{s' \sim p(\cdot|s, \pi_{\hat{Q}}(s))} \left[ V^*(s') - V^{\pi_{\hat{Q}}}(s') \right] \\
&\leq 2\varepsilon \sum_{t=0}^{k-1} \gamma^t + \gamma^k (V_{max} - V_{min}) \\
&\leq \frac{2\varepsilon}{1-\gamma} + \gamma^k (V_{max} - V_{min}),
\end{aligned}
$$

which concludes the first part of the proof. Concerning the second inequality, since $V_{min} \leq Q^*(s, a) \leq V_{max}$, we have

$$
\mathbb{E}_{\hat{Q}} \left[ Q^*(s, \pi_{\hat{Q}}(s)) \right] \geq (1-2\delta)[Q^*(s, \pi^*(s)) - 2\varepsilon] + 2\delta V_{min} \geq Q^*(s, \pi^*(s)) - (2\varepsilon + 2\delta(V_{max} - V_{min})).
$$

Let $\pi_j$ be a policy that replicates $\pi_{\hat{Q}}$ for the first $j$ actions and that is optimal from action $j+1$ onward. We now show by induction that $V^{\pi_j}(s) \geq V^*(s) - \lambda_j$ for all $s$, where $\lambda = \lambda_1 = 2\varepsilon + 2\delta(V_{max} - V_{min})$ and $\lambda_j = \lambda + \gamma\lambda_{j-1}$ for $j > 1$. This clearly holds for $j = 1$:

$$
\begin{aligned}
V^{\pi_1}(s) &= \mathbb{E}_{\hat{Q}} \left[ r(s, \pi_{\hat{Q}}(s)) + \gamma \mathbb{E}_{s' \sim p(\cdot|s, \pi_{\hat{Q}}(s))} [V^*(s')] \right] \\
&= \mathbb{E}_{\hat{Q}} \left[ Q^*(s, \pi_{\hat{Q}}(s)) \right] \\
&\geq Q^*(s, \pi^*(s)) - \lambda \\
&= V^*(s) - \lambda_1.
\end{aligned}
$$

For $j > 1$, assuming that the statement holds for $j - 1$, we have

$$
\begin{aligned}
V^{\pi_j}(s) &= \mathbb{E}_{\hat{Q}} \left[ r(s, \pi_{\hat{Q}}(s)) + \gamma \mathbb{E}_{s' \sim p(\cdot|s, \pi_{\hat{Q}}(s))} [V^{\pi_{j-1}}(s')] \right] \\
&\geq \mathbb{E}_{\hat{Q}} \left[ r(s, \pi_{\hat{Q}}(s)) + \gamma \mathbb{E}_{s' \sim p(\cdot|s, \pi_{\hat{Q}}(s))} [V^*(s') - \lambda_{j-1}] \right] \\
&= \mathbb{E}_{\hat{Q}} \left[ r(s, \pi_{\hat{Q}}(s)) + \gamma \mathbb{E}_{s' \sim p(\cdot|s, \pi_{\hat{Q}}(s))} [V^*(s')] \right] - \gamma\lambda_{j-1} \\
&= \mathbb{E}_{\hat{Q}} \left[ Q^*(s, \pi_{\hat{Q}}(s) \right] - \gamma\lambda_{j-1} \\
&\geq Q^*(s, \pi^*(s)) - \lambda - \gamma\lambda_{j-1} \\
&= V^*(s) - \lambda_j.
\end{aligned}
$$

We now show that $\lim_{j \to \infty} V^{\pi_j}(s) = V^{\pi_{\hat{Q}}}(s)$ for all $s$. Noting that $V^{\pi_j}(s) \geq V^{\pi_{\hat{Q}}}(s)$ (due to the optimality of $\pi_j$ after $j$ steps), we have

$$
\begin{aligned}
0 \leq V^{\pi_j}(s) - V^{\pi_{\hat{Q}}}(s) &= \mathbb{E}_{\tau \sim p_\tau(\cdot|\pi_j, s_0 = s)} \left[ \sum_{i=0}^{\infty} \gamma^i r(s_i, a_i) \right] - \mathbb{E}_{\tau \sim p_\tau(\cdot|\pi_{\hat{Q}}, s_0 = s)} \left[ \sum_{i=0}^{\infty} \gamma^i r(s_i, a_i) \right] \\
&= \mathbb{E}_{\tau \sim p_\tau(\cdot|\pi_j, s_0 = s)} \left[ \sum_{i=j}^{\infty} \gamma^i r(s_i, a_i) \right] - \mathbb{E}_{\tau \sim p_\tau(\cdot|\pi_{\hat{Q}}, s_0 = s)} \left[ \sum_{i=j}^{\infty} \gamma^i r(s_i, a_i) \right] \\
&\leq \sum_{i=j}^{\infty} \gamma^i (R_{max} - R_{min}) \\
&= \gamma^j \frac{R_{max} - R_{min}}{1-\gamma} \xrightarrow[j \to \infty]{} 0.
\end{aligned}
$$

Therefore, by the squeeze theorem, we have $\lim_{j \to \infty} V^{\pi_j}(s) - V^{\pi_{\hat{Q}}}(s) = 0$. Finally, noting that

$$
\lim_{j \to \infty} \lambda_j = \sum_{j=0}^{\infty} \gamma^j \lambda = \frac{\lambda}{1-\gamma},
$$

we have for all $s$:

$$V^{\pi_{\hat{Q}}}(s) = \lim_{j \to \infty} V^{\pi_j}(s) \geq V^*(s) - \lim_{j \to \infty} \lambda_j = V^*(s) - \frac{2\varepsilon + 2\delta(V_{max} - V_{min})}{1 - \gamma}.$$

$\square$

**Lemma 9** (Hoeffding-Serfling Inequalities [4, 33])**.** *Let $\mathcal{X} = \{x_i\}_{i=1}^N$ be a finite set of $N > 1$ real points and $X_n = \{X_j\}_{j=1}^n$ a subset of size $n < N$ sampled uniformly at random without replacement from $\mathcal{X}$. Additionally, denote $\mu = \frac{1}{N}\sum_{i=1}^N x_i$, $a = \min_i x_i$ and $b = \max_i x_i$. Then, for $m = \min(n, N-n)$ and any $\varepsilon > 0$:*

$$\mathbb{P}\left[\left|\frac{1}{n}\sum_{j=1}^n X_j - \mu\right| \geq \varepsilon\right] \leq 2\exp\left\{-\frac{2\varepsilon^2 n}{(1-\frac{n}{N})(1+\frac{1}{m})(b-a)^2}\right\}.$$

*Proof.* First, we show the slightly more general result:

$$1) \qquad \mathbb{P}\left[\frac{1}{n}\sum_{j=1}^n X_j - \mu \geq \varepsilon\right] \leq \exp\left\{-\frac{2\varepsilon^2 n}{(1-\frac{n}{N})(1+\frac{1}{n})(b-a)^2}\right\},$$

$$2) \qquad \mathbb{P}\left[\frac{1}{n}\sum_{j=1}^n X_j - \mu \leq -\varepsilon\right] \leq \exp\left\{-\frac{2\varepsilon^2 n}{(1-\frac{n}{N})(1+\frac{1}{N-n})(b-a)^2}\right\}.$$

The proof of 1) is similar to the one proposed in [4]: Let $Z_n = \frac{1}{n}\sum_{j=1}^n X_j - \mu$. We have for any $\lambda > 0$:

$$\mathbb{P}[Z_n \geq \varepsilon] = \mathbb{P}\left[e^{\lambda n Z_n} \geq e^{\lambda n \varepsilon}\right] \leq \frac{\mathbb{E}[e^{\lambda n Z_n}]}{e^{\lambda n \varepsilon}} \leq \exp\left\{\frac{1}{8}(b-a)^2\lambda^2(n+1)\left(1-\frac{n}{N}\right) - \lambda n \varepsilon\right\},$$

where we have used Markov's inequality along with Proposition 2.3 from [4] (slightly improving the original result proposed by Serfling [33]). The result 1) follows by finding $\lambda$ that minimizes this upper-bound, i.e.,

$$\lambda = \frac{4n\varepsilon}{(n+1)(1-\frac{n}{N})(b-a)^2}.$$

For 2), we note that sampling $X_n$ is equivalent to sampling $X'_{N-n} = \mathcal{X} \setminus X_n$. Therefore:

$$\mathbb{P}\left[\frac{1}{n}\sum_{j=1}^n X_j - \mu \leq -\varepsilon\right] = \mathbb{P}\left[\frac{1}{n}\left(\sum_{j=1}^n X_j - n\mu\right) \leq -\varepsilon\right]$$

$$= \mathbb{P}\left[\frac{1}{n}\left(N\mu - \sum_{j=1}^{N-n} X'_j - n\mu\right) \leq -\varepsilon\right]$$

$$= \mathbb{P}\left[\frac{N-n}{n}\left(\mu - \frac{1}{N-n}\sum_{j=1}^{N-n} X'_j\right) \leq -\varepsilon\right]$$

$$= \mathbb{P}\left[\frac{1}{N-n}\sum_{j=1}^{N-n} X'_j - \mu \geq \frac{n\varepsilon}{N-n}\right]$$

$$\leq \exp\left\{-\frac{2\varepsilon^2 n}{(1-\frac{n}{N})(1+\frac{1}{N-n})(b-a)^2}\right\},$$

where we have used 1) in the last step. Lastly, the final result follows from Boole's inequality (union bound) between 1) and 2). $\square$

### A.2.2 Existence of PAA and AA policies

**Theorem 3** ($\pi_{PAA}$ is $\delta$-$\varepsilon$-PAA). *Let $\mathcal{M}_{\mathcal{I}} = (\mathcal{S}, \mathcal{A}, p, W_q, \mathbf{u}, \gamma)$ be a SMDP with $q \in \mathbb{R}$ and $\hat{p}$ an approximate dynamics model such that $d \triangleq \sup_{(s,a)} D_{KL}(p(\cdot|s,a) \| \hat{p}(\cdot|s,a)) < \frac{\varepsilon^2 (1-\gamma)^6}{8 \Delta U^2}$ for any desired tolerances $\varepsilon > 0$ and $0 < \delta < 1$. For any $k \geq \log_\gamma \left( \frac{(1-\gamma)\varepsilon}{\Delta U} - \frac{\sqrt{8d}}{(1-\gamma)^2} \right)$, define $\beta \triangleq \left( \frac{(1-\gamma)^2 \varepsilon}{8} - \frac{\sqrt{d} \Delta U}{\sqrt{8}(1-\gamma)} - \frac{(1-\gamma)\gamma^k \Delta U}{8} \right)$ and let $\pi_{PAA}$ be the policy defined in Eq. (7) with parameters*

- $H \geq \max \left\{ 1, \log_\gamma \left( \frac{\beta}{\Delta U} \right) \right\}$,

- $K \geq \frac{\Delta U^2}{\beta^2} \left( (H-1) \ln \left( \sqrt[H-1]{24k}(H-1) |\mathcal{A}| \frac{\Delta U^2}{\beta^2} \right) + \ln \left( \frac{1}{\delta} \right) \right)$,

- $C \geq \frac{\gamma^2}{(1-\gamma)^2} K$,

- $n \geq N \left( 1 + \frac{\Delta U^2 N}{2K} \Gamma \left( \beta, U_{min}, U_{max}, q \right) \right)^{-1}$,

*and where $\Gamma \left( \beta, U_{min}, U_{max}, q \right)$ is a function defined in Eq. (9). Then $\pi_{PAA}$ is a $\delta$-$\varepsilon$-PAA policy.*

*Proof.* Similarly to [19], the core idea (and challenge) of the proof is to bound $|Q^*(s,a) - \hat{Q}^H(s,a)|$ for all state-action pairs so that Lemma 7. However, in contrast with [19], we must now deal with an approximate dynamics model, as well as an approximate reward, which significantly complicates the task. To do so, we write the following for $h > 0$ (omitting the dependency of $K$, $C$ and $I_n$ in the notation):

$$
\begin{aligned}
Q^*(s,a) - \hat{Q}^h(s,a) &= r_{\mathcal{I}}(s,a) - \hat{r}_{I_n}(s,a) + \gamma \left( \mathbb{E}_{s' \sim p(\cdot|s,a)}[V^*(s')] - \hat{\mathbb{E}}^C_{s' \sim \hat{p}(\cdot|s,a)}[\hat{V}^{h-1}(s')] \right) \\
&= \mathbb{E}_{s' \sim p(\cdot|s,a)}[W_q(\mathbf{u}(s'); \mathcal{I})] - \hat{\mathbb{E}}^K_{s' \sim \hat{p}(\cdot|s,a)}[W_q(\mathbf{u}(s'); I_n)] \\
&\quad + \gamma(\mathbb{E}_{s' \sim p(\cdot|s,a)}[V^*(s')] - \hat{\mathbb{E}}^C_{s' \sim \hat{p}(\cdot|s,a)}[\hat{V}^{h-1}(s')]) \\
&= \mathbb{E}_{s' \sim p(\cdot|s,a)}[W_q(\mathbf{u}(s'); \mathcal{I})] - \mathbb{E}_{s' \sim p(\cdot|s,a)}[W_q(\mathbf{u}(s'); I_n)] \\
&\quad + \mathbb{E}_{s' \sim p(\cdot|s,a)}[W_q(\mathbf{u}(s'); I_n)] - \mathbb{E}_{s' \sim \hat{p}(\cdot|s,a)}[W_q(\mathbf{u}(s'); I_n)] \\
&\quad + \mathbb{E}_{s' \sim \hat{p}(\cdot|s,a)}[W_q(\mathbf{u}(s'); I_n)] - \hat{\mathbb{E}}^K_{s' \sim \hat{p}(\cdot|s,a)}[W_q(\mathbf{u}(s'); I_n)] \\
&\quad + \gamma(\mathbb{E}_{s' \sim p(\cdot|s,a)}[V^*(s')] - \mathbb{E}_{s' \sim \hat{p}(\cdot|s,a)}[V^*(s')]) \\
&\quad + \gamma(\mathbb{E}_{s' \sim \hat{p}(\cdot|s,a)}[V^*(s')] - \hat{\mathbb{E}}^C_{s' \sim \hat{p}(\cdot|s,a)}[V^*(s')]) \\
&\quad + \gamma(\hat{\mathbb{E}}^C_{s' \sim \hat{p}(\cdot|s,a)}[V^*(s')] - \hat{\mathbb{E}}^C_{s' \sim \hat{p}(\cdot|s,a)}[\hat{V}^{h-1}(s')])
\end{aligned}
$$

such that, for any state-action pair,

$$
\begin{aligned}
|Q^*(s,a) - \hat{Q}^h(s,a)| &\leq \mathbb{E}_{s' \sim p(\cdot|s,a)}[\underbrace{|W_q(\mathbf{u}(s'); \mathcal{I}) - W_q(\mathbf{u}(s'); I_n)|}_{Z_1}] \\
&\quad + \underbrace{|\mathbb{E}_{s' \sim p(\cdot|s,a)}[W_q(\mathbf{u}(s'); I_n)] - \mathbb{E}_{s' \sim \hat{p}(\cdot|s,a)}[W_q(\mathbf{u}(s'); I_n)]|}_{Z_2} \\
&\quad + \underbrace{|\mathbb{E}_{s' \sim \hat{p}(\cdot|s,a)}[W_q(\mathbf{u}(s'); I_n)] - \hat{\mathbb{E}}^K_{s' \sim \hat{p}(\cdot|s,a)}[W_q(\mathbf{u}(s'); I_n)]|}_{Z_3} \\
&\quad + \gamma \underbrace{|\mathbb{E}_{s' \sim p(\cdot|s,a)}[V^*(s')] - \mathbb{E}_{s' \sim \hat{p}(\cdot|s,a)}[V^*(s')]|}_{Z_4} \\
&\quad + \gamma \underbrace{|\mathbb{E}_{s' \sim \hat{p}(\cdot|s,a)}[V^*(s')] - \hat{\mathbb{E}}^C_{s' \sim \hat{p}(\cdot|s,a)}[V^*(s')]|}_{Z_5} \\
&\quad + \gamma \underbrace{\hat{\mathbb{E}}^C_{s' \sim \hat{p}(\cdot|s,a)}[|V^*(s') - \hat{V}^{h-1}(s')|]}_{Z_6}
\end{aligned}
$$

where we have used the triangle inequality, the linearity of expectation and the fact that $|\mathbb{E}[X]| \leq \mathbb{E}[|X|]$ for any random variable $X$. From Lemma 5, we have that $Z_1 \leq \varepsilon_1$ with probability at least $1 - \delta_1$ where

$$\delta_1 \triangleq 2\exp\left(-\frac{2n\varepsilon_1^2}{(1-\frac{n}{N})(1+\frac{1}{m})}\Gamma(\varepsilon_1, U_{min}, U_{max}, q)\right) \leq 2\exp\left(-\frac{n\varepsilon_1^2}{1-\frac{n}{N}}\Gamma(\varepsilon_1, U_{min}, U_{max}, q)\right).$$

Recall that $U_{min} \leq W_q(\mathbf{u}(s)) \leq U_{max}$ and $V_{min} = \frac{U_{min}}{1-\gamma} \leq V^*(s) \leq \frac{U_{max}}{1-\gamma} = V_{max}$ for any state $s$ and utility profile $\mathbf{u}$. Therefore, from Lemma 6, we have (with probability 1)

$$Z_2 \leq 2\Delta U \sqrt{\min\{\frac{d}{2}, 1-e^{-d}\}} \triangleq \varepsilon_2 \qquad \text{and} \qquad Z_4 \leq 2\frac{\Delta U}{1-\gamma}\sqrt{\min\{\frac{d}{2}, 1-e^{-d}\}} \triangleq \varepsilon_4, \quad (16)$$

where $\Delta U \triangleq U_{max} - U_{min}$ and $d \triangleq \sup_{(s,a)} D_{KL}(p(\cdot|s,a)||\hat{p}(\cdot|s,a))$. Furthermore, by the standard Hoeffding's inequality, we have $Z_3 \leq \varepsilon_3$ with probability at least $1 - \delta_3$ where

$$\delta_3 \triangleq 2\exp\left(-\frac{2K\varepsilon_3^2}{\Delta U^2}\right),$$

and $Z_5 \leq \varepsilon_5$ with probability at least $1 - \delta_5$ where

$$\delta_5 \triangleq 2\exp\left(-\frac{2C(1-\gamma)^2\varepsilon_5^2}{\Delta U^2}\right).$$

Finally, we have

$$\hat{\mathbb{E}}^C_{s' \sim \hat{p}(\cdot|s,a)}[|V^*(s') - \hat{V}^{h-1}(s')|] = \frac{1}{C}\sum_{i=1}^{C}|V^*(s_i) - \hat{V}^{h-1}(s_i)|$$

$$= \frac{1}{C}\sum_{i=1}^{C}|\max_a Q^*(s_i, a) - \max_a \hat{Q}^{h-1}(s_i, a)|$$

$$\leq \frac{1}{C}\sum_{i=1}^{C}|Q^*(s_i, \tilde{a}_i) - \hat{Q}^{h-1}(s_i, \tilde{a}_i)|$$

with

$$\tilde{a}_i \triangleq \begin{cases} \arg\max_a Q^*(s_i, a) & \text{if } \max_a Q^*(s_i, a) \geq \max_a \hat{Q}^{h-1}(s_i, a) \\ \arg\max_a \hat{Q}^{h-1}(s_i, a) & \text{if } \max_a Q^*(s_i, a) < \max_a \hat{Q}^{h-1}(s_i, a) \end{cases},$$

and where the last inequality is obtained after a careful analysis of the absolute value operator. This suggests that we can proceed by induction on $h$. Let $\alpha_0 \triangleq \frac{\Delta U}{1-\gamma}$ and $\alpha_h \triangleq \varepsilon_1 + \varepsilon_2 + \varepsilon_3 + \gamma(\varepsilon_4 + \varepsilon_5 + \alpha_{h-1})$ for $h > 0$, and let $\phi_0 \triangleq 0$ and $\phi_h \triangleq \delta_1 + \delta_3 + \delta_5 + C|\mathcal{A}|\phi_{h-1}$ for $h > 0$. We start the induction by noting that, for any state-action pair,

$$|Q^*(s,a) - \hat{Q}^0(s,a)| = |Q^*(s,a)| \leq \frac{\Delta U}{1-\gamma} = \alpha_0 \qquad \text{with probability } 1 = 1 - \phi_0$$

For $h > 1$, assuming that, for any state-action pair, $|Q^*(s,a) - \hat{Q}^{h-1}(s,a)| \leq \alpha_{h-1}$ with probability at least $1 - \phi_{h-1}$, we have from above

$$|Q^*(s,a) - \hat{Q}^h(s,a)| \leq \varepsilon_1 + \varepsilon_2 + \varepsilon_3 + \gamma(\varepsilon_4 + \varepsilon_5 + \alpha_{h-1}) = \alpha_h \qquad (17)$$

with probability at least $1 - \delta_1 - \delta_3 - \delta_5 - C|\mathcal{A}|\phi_{h-1} = 1 - \phi_h$, as we require that all $C$ estimates of $\hat{Q}^{h-1}$ are accurate for each action. Solving for $\alpha_H$, we get

$$\alpha_H = \sum_{i=0}^{H-1}\gamma^i(\varepsilon_1 + \varepsilon_2 + \varepsilon_3 + \gamma(\varepsilon_4 + \varepsilon_5)) + \gamma^H \frac{\Delta U}{1-\gamma}$$

$$= (\varepsilon_1 + \varepsilon_2 + \varepsilon_3 + \gamma(\varepsilon_4 + \varepsilon_5))\frac{1-\gamma^H}{1-\gamma} + \gamma^H \frac{\Delta U}{1-\gamma}$$

$$\leq \frac{\varepsilon_1 + \varepsilon_2 + \varepsilon_3 + \gamma(\varepsilon_4 + \varepsilon_5) + \gamma^H \Delta U}{1-\gamma}. \qquad (18)$$

Note that $\varepsilon_2$ and $\varepsilon_4$ in Eq. (16) are non-controllable and only depend on the accuracy of the environment model $\hat{p}$. If we require $|V^*(s) - V^{\pi_{PAA}}(s)| \le \varepsilon$, then we must have $\frac{2(\varepsilon_2 + \gamma \varepsilon_4)}{(1-\gamma)^2} < \varepsilon$ from Lemma 7. Using the definitions of $\varepsilon_2$ and $\varepsilon_4$ from Eq. (16), this condition becomes $\min\{\frac{d}{2}, 1 - e^{-d}\} < \frac{(1-\gamma)^6 \varepsilon^2}{16\Delta U^2}$ with $d \triangleq \sup_{(s,a)} D_{KL}(p(\cdot|s,a)\|\hat{p}(\cdot|s,a))$. Since $\varepsilon \le \frac{\Delta U}{(1-\gamma)}$ (as the bound would be trivial otherwise), the right-hand side of the inequality is less than 1, and thus it is less restrictive (on $\hat{p}$) to only bound $\frac{d}{2}$ (as $\frac{x}{2} \le 1 - e^{-x}$ for $x \in [0, 1]$), which concludes the proof on the required accuracy of the dynamics model. Now that we have ensured that the environment model is accurate enough, we must choose the algorithm's parameters (namely $K$, $C$ and $n$) to ensure that the other inaccuracies do not exceed the remaining "approximation budget"

$$\varepsilon - \frac{2(\varepsilon_2 + \gamma \varepsilon_4)}{(1-\gamma)^2} \ge \varepsilon - \frac{\sqrt{8d}\Delta U}{(1-\gamma)^3} > 0.$$

Solving for $\phi_H$, we get

$$\begin{aligned}
\phi_H &= (\delta_1 + \delta_3 + \delta_5) \sum_{i=0}^{H-1} (C|\mathcal{A}|)^i \\
&= (\delta_1 + \delta_3 + \delta_5) \frac{(C|\mathcal{A}|)^H - 1}{C|\mathcal{A}| - 1} \\
&\le 2(\delta_1 + \delta_3 + \delta_5)(C|\mathcal{A}|)^{H-1},
\end{aligned} \tag{19}$$

where the last inequality follows from the fact that $C|\mathcal{A}| - 1 \ge \frac{1}{2}C|\mathcal{A}|$ as $C \ge 1$ and $|\mathcal{A}| \ge 2$. Finally, using part 1) of Lemma 7, we get that for any state $s$, with probability at least $1 - 2k\phi_H$, $h \in \mathbb{N}^*$,

$$V^*(s) - V^{\pi_{PAA}}(s) \le \frac{2\alpha_H + \gamma^k \Delta U}{1-\gamma} \tag{20}$$

$$\le \frac{2}{(1-\gamma)^2} \left( \varepsilon_1 + \varepsilon_2 + \varepsilon_3 + \gamma(\varepsilon_4 + \varepsilon_5) + \gamma^H \Delta U + \frac{1-\gamma}{2}\gamma^k \Delta U \right).$$

That is, using the definitions of $\varepsilon_2$ and $\varepsilon_4$, and imposing the tolerance $\varepsilon$, we require

$$\varepsilon_1 + \varepsilon_3 + \gamma \varepsilon_5 + \gamma^H \Delta U + \frac{1-\gamma}{2}\gamma^k \Delta U \le \frac{(1-\gamma)^2}{2} \left( \varepsilon - \frac{\sqrt{8d}\Delta U}{(1-\gamma)^3} \right).$$

We fix $\varepsilon_1 = \varepsilon_3 = \gamma \varepsilon_5 = \gamma^H \Delta U = \frac{(1-\gamma)^2}{8} \left( \varepsilon - \frac{\sqrt{8d}\Delta U}{(1-\gamma)^3} - \frac{1}{1-\gamma}\gamma^k \Delta U \right) \triangleq \beta$. From that, we directly obtain the required planning "depth" $H = \max\left\{ 1, \left\lceil \log_\gamma \left( \frac{\beta}{\Delta U} \right) \right\rceil \right\}$, as well as the lower bound $k \ge \log_\gamma \left( \frac{(1-\gamma)\varepsilon}{\Delta U} - \frac{\sqrt{8d}}{(1-\gamma)^2} \right)$. Additionally, we choose $C = \frac{\gamma^2}{(1-\gamma)^2}K$ such that $\delta_3 = \delta_5 = 2\exp\left( -\frac{2K\beta^2}{(\Delta U)^2} \right)$, and we choose $n$ such that $\delta_1 \le \delta_3$, or equivalently

$$\frac{n\Gamma(\beta, U_{min}, U_{max}, q)}{(1 - \frac{n}{N})} \ge \frac{2K}{\Delta U^2}.$$

This is satisfied for

$$n \ge N \left( 1 + \frac{\Delta U^2 \Gamma(\beta, U_{min}, U_{max}, q)N}{2K} \right)^{-1}.$$

With this choice of $C$ and $n$, we can write

$$\delta_1 + \delta_3 + \delta_5 \le 6\exp\left( -\frac{2K\beta^2}{\Delta U^2} \right).$$

The last step is to choose $K$ such that $2k\phi_H \le \delta$, that is

$$24k(K|\mathcal{A}|)^{H-1} \exp\left( -\frac{2K\beta^2}{\Delta U^2} \right) \le \delta. \tag{21}$$

If $H = 1$ (i.e., $\gamma \leq \frac{\beta}{\Delta U}$), we can simply choose

$$K \geq \frac{\Delta U^2}{2\beta^2} \ln\left(\frac{24k}{\delta}\right),$$

and if $H > 1$ (i.e., $\gamma > \frac{\beta}{\Delta U}$), we can choose

$$K \geq \frac{\Delta U^2}{\beta^2}\left((H-1)\ln\left(\sqrt[H-1]{24k}(H-1)|\mathcal{A}|\frac{\Delta U^2}{\beta^2}\right) + \ln\left(\frac{1}{\delta}\right)\right).$$

Indeed, setting $x \triangleq \sqrt[H-1]{24k}(H-1)|\mathcal{A}|$ and $y \triangleq \frac{\beta}{\Delta U}$ and substituting the expression for $K$, the left-hand term of inequality (21) can be rewritten as

$$\left(\frac{x}{y^2}\right)^{H-1}\left(\ln\left(\frac{x}{y^2}\right) + \frac{1}{H-1}\ln\left(\frac{1}{\delta}\right)\right)^{H-1}\left(\frac{y^2}{x}\right)^{2(H-1)}\delta^2 = \left(\frac{\ln\left(\frac{x}{y^2\delta^{\frac{1}{H-1}}}\right)}{\frac{x}{y^2}}\right)^{H-1}\delta^2$$

$$= \left(\frac{\ln\left(\frac{x}{y^2\delta^{\frac{1}{H-1}}}\right)}{\frac{x\delta^{\frac{1}{H-1}}}{y^2\delta^{\frac{1}{H-1}}}}\right)^{H-1}\delta^2$$

$$= \left(\frac{\ln\left(\frac{x}{y^2\delta^{\frac{1}{H-1}}}\right)}{\frac{x}{y^2\delta^{\frac{1}{H-1}}}}\right)^{H-1}\delta$$

$$\leq \delta, \tag{22}$$

where we have used the observation that $x \geq 1$ and $y \leq 1$ along with the fact that $0 \leq \frac{\ln(z)}{z} \leq 1$ for all $z \geq 1$ in the last step. Finally, assuming there exists an optimal policy in $\mathcal{M}_\mathcal{I}$ (see [12, 37] for the necessary conditions), and knowing from Lemma 1 that $V^\pi(s) = \mathcal{W}^\pi(s)$ for any policy, we know that $V^*(s) - V^{\pi_{PAA}}(s) \leq \varepsilon$ implies $\mathcal{W}^{\pi_{PAA}}(s) \geq \sup_{\pi'}\mathcal{W}^{\pi'}(s) - \varepsilon$. $\qquad\square$

**Corollary 4.** *Let $\mathcal{M}_\mathcal{I} = (\mathcal{S}, \mathcal{A}, p, W_q, \mathbf{u}, \gamma)$ be a SMDP with $q \in \mathbb{R}$ and $\hat{p}$ an approximate dynamics model such that $d \triangleq \sup_{(s,a)} D_{KL}(p(\cdot|s,a)\|\hat{p}(\cdot|s,a)) < \frac{\varepsilon^2(1-\gamma)^6}{8\Delta U^2}$ for any desired tolerance $\varepsilon > 0$. Define $\beta \triangleq \frac{(1-\gamma)^2\varepsilon}{10} - \frac{\sqrt{2d}\Delta U}{5(1-\gamma)}$ and let $\pi_{PAA}$ be the policy defined in Eq. (7) with parameters $H, C$ and $n$ as in Theorem 3 and $K \geq \frac{\Delta U^2}{\beta^2}\left((H-1)\ln\left(\sqrt[H-1]{12}(H-1)|\mathcal{A}|\frac{\Delta U^2}{\beta^2}\right) + \ln\left(\frac{\Delta U}{\beta}\right)\right)$. Then $\pi_{PAA}$ is an $\varepsilon$-AA policy.*

*Proof.* The first part of the proof is identical to Theorem 3, but we use part 2) of Lemma 7 instead of part 1) in Eq. (20). With that, we have for any state $s$:

$$V^*(s) - V^{\pi_{PAA}}(s) \leq \frac{2(1-\gamma)\alpha_H + 2\phi_H\Delta U}{(1-\gamma)^2}$$

$$\leq \frac{2}{(1-\gamma)^2}\left(\varepsilon_1 + \varepsilon_2 + \varepsilon_3 + \gamma(\varepsilon_4 + \varepsilon_5) + \gamma^H\Delta U + 2\Delta U(\delta_1 + \delta_3 + \delta_5)(C|\mathcal{A}|)^{H-1}\right).$$

That is, using the definitions of $\varepsilon_2$ and $\varepsilon_4$, and imposing the tolerance $\varepsilon$, we require

$$\varepsilon_1 + \varepsilon_3 + \gamma\varepsilon_5 + \gamma^H\Delta U + 2\Delta U(\delta_1 + \delta_3 + \delta_5)(C|\mathcal{A}|)^{H-1} \leq \frac{(1-\gamma)^2}{2}\left(\varepsilon - \frac{\sqrt{8d}\Delta U}{(1-\gamma)^3}\right).$$

We fix $\varepsilon_1 = \varepsilon_3 = \gamma\varepsilon_5 = \gamma^H\Delta U = \beta \triangleq \frac{(1-\gamma)^2}{10}\left(\varepsilon - \frac{\sqrt{8d}\Delta U}{(1-\gamma)^3}\right)$. From that, we directly obtain the required planning "depth" $H = \max\left\{1, \left\lceil\log_\gamma\left(\frac{\beta}{\Delta U}\right)\right\rceil\right\}$. Similarly to Theorem 3, we choose $C = \frac{\gamma^2}{(1-\gamma)^2}K$ and

$$n \geq N\left(1 + \frac{\Delta U^2\Gamma(\beta, U_{min}, U_{max}, q)N}{2K}\right)^{-1},$$

such that

$$\delta_1 + \delta_3 + \delta_5 \le 6 \exp\left(-\frac{2K\beta^2}{\Delta U^2}\right).$$

The last step is to choose $K$ such that

$$12(K|\mathcal{A}|)^{H-1} \exp\left(-\frac{2K\beta^2}{\Delta U^2}\right) \le \frac{\beta}{\Delta U}. \tag{23}$$

If $H = 1$ (i.e., $\gamma \le \frac{\beta}{\Delta U}$), we can simply choose

$$K \ge \frac{\Delta U^2}{2\beta^2} \ln\left(\frac{12\Delta U}{\beta}\right),$$

and if $H > 1$ (i.e., $\gamma > \frac{\beta}{\Delta U}$), we can choose

$$K \ge \frac{\Delta U^2}{\beta^2}\left((H-1)\ln\left(\sqrt[H-1]{12}(H-1)|\mathcal{A}|\frac{\Delta U^2}{\beta^2}\right) + \ln\left(\frac{\Delta U}{\beta}\right)\right).$$

We show that this choice of $K$ satisfies Eq. (23) by setting $\delta = \frac{\beta}{\Delta U}$ and $k = \frac{1}{2}$ in Eq. (22). We also conclude the proof using Lemma 1. $\qquad\square$

### A.2.3  Safe policies

**Theorem 8** (Safeguarding a Black-Box Policy). *Given a SMDP $\mathcal{M}_\mathcal{I} = (\mathcal{S}, \mathcal{A}, p, W_q, \mathbf{u}, \gamma)$ with $q \in \mathbb{R}$, a predictive model $\hat{p}$ and desired tolerances $\omega \in [\mathcal{W}_{min}, \mathcal{W}_{max}]$ and $0 < \delta < 1$, define $\hat{Q}_\omega(s, a) \triangleq \hat{Q}^H(s, a; K, C, I_n)$ with $\hat{Q}^H$ given in Eq. (6) and any $H, K, C, n \ge 1$. For any policy $\pi$, let $\pi_{safe}$ be the restricted version of $\pi$ obtained with the restricted subsets $\mathcal{A}_{safe}(s) \triangleq \{a \in \mathcal{A} : \hat{Q}_\omega(s, a) \ge \gamma\omega + U_{max} + \alpha\}$, where*

$$\alpha \triangleq \frac{2\Delta U d'}{(1-\gamma)^2} + \frac{\sqrt{\ln\left(\frac{12(C|\mathcal{A}|)^{H-1}}{\delta}\right)}}{1-\gamma}\left(\sqrt{\frac{N-n}{nN\Gamma_{max}}} + \sqrt{\frac{\Delta U^2}{2K}} + \gamma\sqrt{\frac{\Delta U^2}{2C(1-\gamma)^2}}\right) + \frac{\gamma^H \Delta U}{1-\gamma},$$

*and with the shortened notation $\Gamma_{max} \triangleq \Gamma(U_{max}, U_{min}, U_{max}, q)$, $d' \triangleq \sqrt{\min\{\frac{d}{2}, 1 - e^{-d}\}}$ and $d \triangleq \sup_{(s,a)} D_{KL}(p(\cdot|s,a)\|\hat{p}(\cdot|s,a))$. Then $\pi_{safe}$ is $\delta$-$\omega$-safe.*

*Proof.* Assuming there exists an optimal policy in $\mathcal{M}_\mathcal{I}$ (see [12, 37] for the necessary conditions), and knowing from Lemma 1 that $V^\pi(s) = \mathcal{W}^\pi(s)$ for any policy, we can write $\sup_{\pi'} \mathcal{W}^{\pi'}(s) = V^*(s)$. Therefore, the condition for safe actions becomes $\mathbb{E}_{s'\sim p(\cdot|s,a)}[V^*(s')] \ge \omega$. Using the definition of the optimal state-action value function $Q^*$:

$$Q^*(s, a) = r_\mathcal{I}(s, a) + \gamma\mathbb{E}_{s'\sim p(\cdot|s,a)}[V^*(s')],$$

we can rewrite this condition once again as

$$Q^*(s, a) \ge \gamma\omega + r_\mathcal{I}(s, a).$$

We know from Eq. (17) in the proof of Theorem 3 that, for any state-action pair $(s, a)$, $|Q^*(s, a) - \hat{Q}^H(s, a; K, C, I_n)| \le \alpha_H$ with probability at least $1 - \phi_H$, where $\alpha_H$ and $\phi_H$ are given in Eq. (18) and (19), respectively. That is, if $\hat{Q}^H(s, a; K, C, I_n) \ge \gamma\omega + r_\mathcal{I}(s, a) + \alpha_H$, then the above condition is satisfied with probability at least $1 - \phi_H$.

From Eq. (18), we have

$$\alpha_H \le \frac{\varepsilon_2 + \gamma\varepsilon_4}{1-\gamma} + \frac{\varepsilon_1 + \varepsilon_3 + \gamma\varepsilon_5}{1-\gamma} + \frac{\gamma^H \Delta U}{1-\gamma}, \tag{24}$$

where (using $d \triangleq \sup_{(s,a)} D_{KL}(p(\cdot|s,a)\|\hat{p}(\cdot|s,a)))$:

$$\varepsilon_1 = \sqrt{\frac{N-n}{nN\Gamma(\varepsilon_1, U_{min}, U_{max}, q)} \ln\left(\frac{2}{\delta_1}\right)} \leq \sqrt{\frac{N-n}{nN\Gamma(U_{max}, U_{min}, U_{max}, q)} \ln\left(\frac{2}{\delta_1}\right)},$$

$$\varepsilon_2 = 2\Delta U \sqrt{\min\{\frac{d}{2}, 1 - e^{-d}\}},$$

$$\varepsilon_3 = \sqrt{\frac{\Delta U^2}{2K} \ln\left(\frac{2}{\delta_3}\right)},$$

$$\varepsilon_4 = \frac{2\Delta U}{1-\gamma} \sqrt{\min\{\frac{d}{2}, 1 - e^{-d}\}},$$

$$\varepsilon_5 = \sqrt{\frac{\Delta U^2}{2C(1-\gamma)^2} \ln\left(\frac{2}{\delta_5}\right)}.$$

Finally, in order to obtain a $\delta$-$\omega$-safe policy, we must have $\phi_H \leq \delta$. From Eq. (19), this is satisfied if

$$2(\delta_1 + \delta_3 + \delta_5)(C|\mathcal{A}|)^{H-1} \leq \delta,$$

or similarly if $\delta_1 = \delta_3 = \delta_5 = \frac{\delta}{6(C|\mathcal{A}|)^{H-1}}$.

Defining $d' = \sqrt{\min\{\frac{d}{2}, 1 - e^{-d}\}}$ and $\Gamma_{max} = \Gamma(U_{max}, U_{min}, U_{max}, q)$, and substituting $\varepsilon_1, \varepsilon_2, \varepsilon_3, \varepsilon_4, \varepsilon_5, \delta_1, \delta_3, \delta_5$ in Eq. (24), we obtain

$$\alpha_H \leq \frac{2\Delta U d'}{(1-\gamma)^2} + \frac{\sqrt{\ln\left(\frac{12(C|\mathcal{A}|)^{H-1}}{\delta}\right)}}{1-\gamma}\left(\sqrt{\frac{N-n}{nN\Gamma_{max}}} + \sqrt{\frac{\Delta U^2}{2K}} + \gamma\sqrt{\frac{\Delta U^2}{2C(1-\gamma)^2}}\right) + \frac{\gamma^H \Delta U}{1-\gamma}.$$

Using the fact that $r_{\mathcal{I}}(s,a) \leq U_{max}$ for any state-action pair, we can conservatively define the restricted subsets of safe actions as $\mathcal{A}_{safe}(s) = \{a : \hat{Q}^H(s,a; K, C, I_n) \geq \gamma\omega + U_{max} + \alpha\}$ for any state $s$, where $\alpha$ is the right-hand term of the above inequality. Therefore, restricting any policy $\pi$ with these subsets ensures that $\mathbb{E}_{s' \sim p(\cdot|s,a)}[V^*(s')] \geq \omega$ with probability at least $1 - \delta$ for any action $a$ that has a non-zero probability of being selected. $\qquad\square$

