# OpenReview forum: "Can an AI Agent Safely Run a Government? Existence of Probably Approximately Aligned Policies"
_NeurIPS.cc/2024/Conference — NeurIPS 2024 poster_

### Official Review · Reviewer_nJtj · 2024-07-09

**Soundness:** 4
**Presentation:** 4
**Contribution:** 3
**Rating:** 7
**Confidence:** 2

**Summary:**

This paper defines social markov decision processes (SMDPs) as an MDP generalization incorporating a population of individuals with distinct utility profiles aggregated by a social welfare function. It provides a novel quantitative definition of alignment in this context, then leverages this definition to characterize probably approximately aligned policies and safe policies, prove the conditions under which they exist, and relate them to the accuracy of the reward model.

**Strengths:**

1. This paper is well written, and the background is particularly clear.
2. The definitions and theoretical results are thorough and rigorous. This paper precisely relates the probability of aligned behavior to the world model accuracy, which I believe is valuable.
3. This paper acknowledges that realistic inaccuracy in the world model could cause intolerable uncertainty in the PAA policy, and shows a more practical approach (safeguarding a black-box policy).

**Weaknesses:**

Even the more practical approach of safeguarding a black-box policy may have severe limitations. I believe the paper would be strengthened by a discussion of the feasibility of this -- in particular, what is computational complexity of computing $\mathcal{A}_{safe}$ for a SMDP?

Typo: On line 277, I believe "expansive" should be "expensive".

**Questions:**

How does the SMDP formalism handle individuals that give assessments on different scales? What assumption(s) does it rely on regarding interpersonal comparisons of utility?

**Limitations:**

This paper includes an excellent discussion of the limitations of these results, including the theoretical conditions under which PAA and safe policies will be unreliable. The paper also discusses further practical and philosophical limitations in Section 5.

---

> ### Author Rebuttal · Authors · 2024-08-06
>
> We thank the reviewer for their comments and for the time spent reviewing our paper.
>
> **Feasibility of safeguarding black-box policies in SMDPs**
>
> We argue that the main issue is not whether safeguarding a black-box policy is feasible, but whether it produces a useful policy. Indeed, the practicality of the safeguarding method comes from the fact that the computational complexity of finding $\mathcal A_{safe}$ can be made as small as desired based on the resources available (with $K$, $C$, $H$, and $n$ being chosen as desired, unlike the PAA policy where they are determined by $\epsilon$, $\delta$, $\gamma$ and $D_{KL}(p\Vert\hat{p})$). However, as these values decrease, the $\alpha$ term in the definition of  $\mathcal A_{safe}$ increases, and the resulting policy may become less useful, as only actions with very high Q-values will be considered safe. Such actions might not even exist (empty $\mathcal{A}_{safe}$), in which case the safe policy is essentially useless. On the other hand, deriving the computational complexity to ensure usefulness (i.e., the safe policy always selects an action) cannot be done in the general case, as it would require knowledge of the Q-values for every state-action pairs.
>
> **Scale of the reported utilities**
>
> We believe it is crucial that assessors are instructed to report their utilities for different states within a unified range, such as between 1 and 10. If this range is not standardized, utilities would need to be artificially mapped to a common scale to be aggregated, which we believe would introduce more distortion compared to when this mapping is done internally by the assessor before reporting. However, even with a unified range, interpersonal comparability is not guaranteed, as any given rating within that range might reflect different levels of welfare across different individuals. This issue is philosophical in nature, and there is no feasible experiment to determine whether utilities are truly comparable.
>
> From this point, two approaches can be taken:
>
> - One can simply treat the utilities as if they were comparable, and construct a social welfare function based on them.
> - Alternatively, one can take additional steps to calibrate the utilities using anchor states that should elicit similar welfare across individuals. For instance, in an economic context, assessors could be instructed that a rating of 9 corresponds to landing their dream job, while a rating of 2 corresponds to being unemployed. Although this may not necessarily make utilities more comparable (as people may have different sensibilities to these events), it does provide a more interpretable framework.
>
> In general, while interpersonal comparability might seem like a strong assumption, there are practical examples where it appears evident. For instance, it is generally conceivable that two people can determine who enjoys chocolate more through a conversation. This necessarily implies that their utilities have a certain level of comparability.

---

> > ### Comment · Reviewer_nJtj · 2024-08-09
> >
> > Thank you for the nuanced discussion. This work seems thorough to me, and I will maintain my original rating.

---

### Official Review · Reviewer_KXy4 · 2024-07-12

**Soundness:** 3
**Presentation:** 3
**Contribution:** 3
**Rating:** 6
**Confidence:** 2

**Summary:**

This paper applies ideas from the Probably Approximately Correct framework to agent alignment. The paper defines a new idea of a policy which is Probably Approximately Aligned and explores the existence of such policies under certain assumptions of social welfare and models of the world. The authors show that probably approximately aligned (and approximately aligned) policies exist when there is a sufficiently accurate world model. However, to compute this policy is quite expensive. Thus, the authors also develop the idea of a safe policy which can be derived using a PAA policy and seems to be a policy that will probably not result in a catastrophically bad state.

**Strengths:**

Overall the paper appears to be a very reasonable application of a well established form of analysis into a novel domain.

The main idea of providing bounds for the quality of an agent's policy is very important and will likely be the focus of much work in the near future. This is quite useful work and appears to me as the potential basis for work that can eventually have significant beneficial impact on the world.

The paper is generally well written and the motivation is clear. In places the math is a little dense but it seems to be as approachable as it can be for this sort of analysis. I do certainly appreciate that you've put a moderate amount of the work into the actual paper rather than stuffing all the important stuff into the appendix.

**Weaknesses:**

Not a weakness, but my disclaimer: I was not able to thoroughly review every detail of the math due to time constraints so my understanding of the paper is limited.

The primary (and minor) issue I see with the paper is that it is quite abstract and doesn't give a clear idea of how close this is to being useful. While obviously difficult to fit into a conference paper, an experimental section may give some intuition for details such as how accurate a world model really needs to be, how beneficial PAA/safe policies are, etc.

It seems that Sec 3.2 is constructive in a sense and provides a PAA policy. Some further commentary on the practicality of this policy (is it entirely impractical to use it for synthetic experiments, or simply impractical in any useful setting/world model?) would help to contextualize the paper.

**Questions:**

You do a good job of stating weaknesses but it seems that the first weakness listed may be quite significant. Is this work essentially just pushing the real difficulty of aligned policies into the task of building a statistically sound world model?

**Limitations:**

Limitations are well stated.

---

> ### Author Rebuttal · Authors · 2024-08-06
>
> We thank the reviewer for their comments and for the time spent reviewing our paper.
>
> **Usefulness of the theoretical results**
>
> We understand that the reviewer's primary concern is the lack of clarity regarding the applicability of the theoretical results to real-world scenarios. While we do not expect PAA (or safe) policies to be implemented in practical applications in the immediate future, we believe this work lays a foundation for future research, where relaxing some constraints could make the approach more practical. For instance, as mentioned in our response to review Fxrx, one potential relaxation is using surrogate models to provide the necessary feedback for constructing the alignment metric. Another relaxation to address the issue of world model accuracy is to develop world models biased towards predicting catastrophic state-action pairs, as errors in predicting non-critical outcomes are less problematic. The rationale behind the paper's fully theoretical approach is to emphasize the existence of a priori alignement (through PAA policies) rather than presenting and testing a specific policy, with the hope that it will spark further research in this direction.
>
> **Practicality of the proposed PAA policy**
>
> Regarding the practicality of the PAA policy introduced in the paper, it is designed primarily to facilitate mathematical proof of its PAA property in a general setting (hence demonstrating the existence of PAA policies in any context). The policy essentially employs a brute force approach by testing all possible actions to identify the best one. However, in a real-world scenario, domain expertise and other simplifying assumptions would certainly be leveraged to eliminate certain state-action pairs, significantly reducing the algorithm's complexity. We intentionally omitted an experimental section to avoid giving the impression that one of the main contribution of the paper is the PAA policy itself, and that it should be implemented as is.
>
> **Are we simply shifting the difficulty to building better world models?**
>
> The main question of the reviewer is a very important one. We address it by exploring two key implications of our work:
>
> - Firstly, as mentioned in the paper’s introduction, *a priori* alignment is feasible only if we have a sufficient understanding of the consequences of each action available to the autonomous agent. However, our results offer an alternative perspective: they indicate that critical actions should not be delegated to an autonomous agent if the available world model is poor when it comes to predicting their effects. In that sense, our work is the first to provide a quantitative framework for determining which actions can be safely entrusted to an autonomous agent. We believe this type of analysis alone could initiate several research threads in the near future.
> - Secondly, while it is true that creating a statistically accurate world model can indeed be as challenging as developing an aligned AI agent, achieving consensus on what constitutes a good world model (i.e., one with high empirical accuracy) is generally more straightforward than defining what constitutes an aligned AI agent (i.e., one with sufficient alignment, which is typically poorly defined). In other words, while our work may shift the challenge to another complex problem, it is one where the solution is easier to verify and where the objective is better defined.

---

> > ### Comment · Reviewer_KXy4 · 2024-08-08
> >
> > Thank you for the well reasoned response. I'm not fully convinced that accurately modeling the world is as easy as you make it out to be but getting into deeper detail is likely out of scope.

---

> > > ### Author Response · Authors · 2024-08-12
> > >
> > > We want to clarify that we are not claiming that *modeling* the world is easy in any general setting. If this impression is conveyed in the paper or the rebuttal, please let us know where, and we will address it. Our intention was to answer your initial question by providing context regarding the implications of our work.
> > >
> > > One alternative approach to understanding these implications is the comparison with LLM alignment. Pre-training a LLM on a large corpus of text essentially builds a world "language" model, where, given a state (the start of a sentence), the model predicts the probability of the next token with little to no consideration of the alignment of the full sentence. Then, an algorithm (e.g., RLHF or DPO) is used to align the model with the user's interests. While the analogy is somewhat limited, our work would correspond to this second phase.

---

### Official Review · Reviewer_Fxrx · 2024-07-13

**Soundness:** 3
**Presentation:** 4
**Contribution:** 3
**Rating:** 6
**Confidence:** 3

**Summary:**

The paper aims to define alignment quantitatively and ensure AI agents' actions are predictable and safe. The paper start by outlines the basics of utility and social choice theory, focusing on quantifying social satisfaction and the conditions under which it is measurable. Next, the paper defines probably approximately aligned (PAA) and approximately aligned (AA) policies and provides a modified sparse sampling algorithm to achieve these policies under certain conditions. The paper also presents the idea of "safe policies" and a method to ensure AI actions are verifiably safe for society.

**Strengths:**

- Originality: This paper introduces a novel, quantitative definition of alignment in social decision-making contexts, drawing from utility and social choice theory.

- Quality: The paper primarily focuses on theoretical contributions rather than empirical experiments. It is well-structured.

- Clarity: The paper provides detailed mathematical derivations and proofs to support the existence of PAA and safe policies. It includes extensive references and context, including foundational works in utility theory, social choice, AI safety, and reinforcement learning, emphasizing the interdisciplinary nature of aligning AI with human values.

- Significance: This work has a significant impact. While primarily theoretical, the work aims to provide a foundation for developing AI systems that could be safely used in critical applications like social governance, policy-making, or resource allocation.

**Weaknesses:**

The safeguarding method is described in a general context, with limited discussion of its applicability to specific real-world problems. Consider adding examples of real-world applications where the safeguarding method could be particularly beneficial. For instance, discuss its application in autonomous vehicle systems, healthcare decision-making, or financial trading algorithms.

**Questions:**

Could you provide more detailed steps on how the safeguarding method can be practically implemented in real-world systems? Consider add roadmap with examples on how to adapt a black-box policy into a safe policy.

**Limitations:**

The authors discuss various limitations of their approach, including computational complexity for large state spaces and strong assumptions about the availability and accuracy of information. The paper also highlights challenges in building reliable world models and the philosophical questions surrounding the informational basis of utilities.

---

> ### Author Rebuttal · Authors · 2024-08-06
>
> We thank the reviewer for their comments and for the time spent reviewing our paper.
>
> We understand that that the primary concern of the reviewer is the gap between the theoretical result presented in the paper and their practical implementation in real-world scenarios (in particular for safe policies). To address this concern, please find below a detailed discussion about the implementability of these policies in several applications, along with a general roadmap.
>
> - **Healthcare decision making**:
> We consider a personalized autonomous doctor (black-box AI agent) for a specific patient. Each week, based on the patient’s health data, the agent determines the appropriate medication for the patient by consulting the patient about various alternatives about aspects such as tolerance to side effects, budget constraints, and preferred timing of administration. In that scenario, the world model could simply be the rate of success and of side effects associated with each medication/dosage during the trial phase of that treatment. Based on these factors, the patient rates each medication alternative on a scale from 1 to 10, and can decide on a minimum discounted welfare level $\omega$. If the agent cannot find a verifiably safe action among its proposed alternatives, it halts and refers the patient to a human doctor. Based on the given guarantees, there will still exist (with high probability) a medication path that generate a future discounted welfare of at lease $\omega$.
> A certified doctor (or several) could also be directly involved in the process. This involvement could take two forms: (i) the doctor acts as a stakeholder accountable for the AI's recommendation, providing feedback on the risks associated with each alternative (also on a scale from 1 to 10 , which is then aggregated with the patient’s score to form a global welfare score), or (ii) the doctor serves as a world model, predicting the impact of each alternative on the patient's health in the absence of other predictive models.
>
> - **Financial trading**:
> We consider the example of an autonomous mutual fund, where investors pool their money to generate returns. The investors could collectively set an aggregate tolerance threshold, $\omega$, that the autonomous trading system must guarantee. Before executing any trade, the system could present various trading strategies along with their associated risks to a subset of the investors. These investors would then assign a utility score (e.g., from 1 to 10) to each option, considering factors like risk and investment type. If no strategy meets the risk tolerance $\omega$, the autonomous trader would abort the transaction and either defer to a human trader or return the funds to the investors. This mechanism ensures that each investor can trust the autonomous agent with their capital, offering a level of verifiable security not always present with human traders. This type of framework could strongly appeal to ethical investors, who prioritize not only the financial returns of their investments but also their moral implications.
>
> - **Autonomous driving**:
> In such a scenario, where decisions need to be made rapidly (likely at the sensor acquisition rate), direct human feedback is typically not feasible. However, cars could communicate with each other. For instance, when a car intends to change its trajectory (such as overtaking, turning, stopping, or accelerating), it could broadcast various proposed paths to nearby vehicles. These neighboring cars would then assess the risk (e.g., collision potential) of each trajectory based on their own planned actions and surroundings (which might not be visible to the initial car). The initial car would then select a safe trajectory or abort the maneuver and return control to the driver if no safe options are available. Although this may not be the most natural application of PAA policies, we believe that designers of autonomous vehicles could benefit from approaching the autonomous driving problem through a social choice framework, where each car would act as a stakeholder within the environment (i.e., the roads).
>
> - **General roadmap**:
> In general, to construct a safe policy from a black-box policy, several factors must be considered. First, we need to establish acceptable tolerance levels, $\omega$ and $\delta$. Next, we must select values for $H, K, C, n$ based on the available resource constraints, such as the number of feedback instances we can gather and the frequency with which we can query the world model to plan each action. Increasing these hyperparameters (excluding $\delta$) typically leads to a lower $\alpha$, resulting in a safe policy that is less prone to delegating its decision authority (i.e., that is more useful). The accuracy of the world model also influences $\alpha$, with a higher accuracy producing a more useful safe policy. If the resulting safe policy is not useful enough, it suggests that either the resources allocated are inadequate or the world model lacks sufficient accuracy  (i.e., $\alpha$ is too large). In both cases, it indicates that we are not yet ready to safely use the black-box policy for that particular application. However, if the amount of feedback required is the bottleneck, one can also imagine a proxy setting where each stakeholder delegates it’s ability to provide feedback to a personalized model (similar to a reward model) that can automatically provide feedback on a given state without human intervention. Note that the safety guarantees would be valid on the alignement metric computed with these surrogate models rather than with the real utilities.

---

> > ### Comment · Reviewer_Fxrx · 2024-08-12
> >
> > Thanks for the response. It was a good discussion.

---

### Official Review · Reviewer_1bQj · 2024-07-23

**Soundness:** 2
**Presentation:** 1
**Contribution:** 2
**Rating:** 5
**Confidence:** 2

**Summary:**

The paper investigates the potential for AI agents to safely make critical decisions, such as those in a government setting, by examining the concept of alignment. It introduces Probably Approximately Aligned (PAA) policies, which are policies that are nearly optimal in aligning with social welfare objectives. The authors draw from utility and social choice theories to provide a quantitative definition of alignment and propose methods to ensure AI actions are verifiably safe for society. They also discuss the practical challenges in implementing such policies and suggest future directions for research in this area. The focus is on developing a theoretical framework that could eventually be applied to AI governance and decision-making processes.

**Strengths:**

The authors draw from utility and social choice theories to provide a quantitative definition of alignment and propose methods to ensure AI actions are verifiably safe for society.

**Weaknesses:**

I think the problem is not well presented.

**Questions:**

I think the problem is not well presented.

E.g.

Section 3.2 - Algorithm for Computing the Policy:
How can you the result of Equation (7) are not clearly explained.

Estimation of Reward (Equation 3):
Equation (3) still appears to be a posterior approach.

**Limitations:**

Yes.

---

> ### Author Rebuttal · Authors · 2024-08-06
>
> We thank the reviewer for their comments and for the time spent reviewing our paper.
>
> It is primarily mentioned that the problem is not well presented. We agree that clarity in presentation is crucial and would greatly appreciate specific suggestions on how to improve it. Below is a detailed discussion about the two specific examples that are put forward:
>
> - **Equation (7)**: To prove our main contribution, the existence of PAA policies (Theorem 2), we construct such a policy in Section 3.2. This policy is built on a near-optimal planning algorithm that approximates Q-values using a world model. Once these Q-values are obtained, the policy selects actions with the highest Q-value, similar to many common reinforcement learning algorithms. Equation (7) formally defines this policy. The technical core of the paper demonstrates that this policy is indeed PAA. In Section 3.2 (with full proofs in the Appendix), we show that the approximate Q-values can be made arbitrarily close to the optimal Q-values. While this specific policy is primarily designed to be provably PAA across various applications, we acknowledge that is is one of many possible PAA policies and we anticipate that future work may develop more efficient algorithms for specific applications.
> - **Equation (3)**: This is still a prior approach according to our definitions because the future states $s'$ are sampled based on the world model $\hat{p}$ rather than the true environment $p$. Thus, the agent does not need to perform actions in the real world to learn the optimal actions. Instead, it can plan according to $\hat{p}$ to ensure that its next action is quasi-optimal with high probability. The policy is therefore a priori aligned, as it ensures alignment before any real actions are taken.
>
> While the main concerns of the reviewer focus on presentation and clarity, we would greatly appreciate a more detailed feedback related to the paper's soundness and contribution, which would help us address the concerns leading to the borderline score. In particular, we would like to understand if the reviewer found that some claims could be better supported (soundness), and if these claims lack originality (or significance) with respect to existing literature (contribution).

---

### Comment · Area_Chair_EzdC · 2024-08-08
**Reviewer - Author Discussion**

Thanks everyone for their hard work on the papers, reviews, and rebuttals. We now have a comprehensive rebuttal from the authors which responds both overall and to each review.

I'd please ask the reviewers to please post a comment acknowledging that they have read the response and ask any followup questions (if any).

This period is to be a discussion between authors and reviewers (Aug 7 - Aug 13) so please do engage now, early in the window, so there is time for a back and forth.

Thanks!

---

### Decision · Program_Chairs · 2024-09-25

**Decision:**

Accept (poster)

**Comment:**

After reviews and a rebuttal/discussion phase that involved all reviewers we have decided that this paper should be accepted to NeurIPS. The reviewers were in agreement that the paper is well written, brings metrics to a novel domain, and is theoretically rigorous.